# Nutritional, molecular, and functional properties of a novel enzymatically hydrolyzed porcine plasma product

Marc Solà-Ginés[1]*, Lluïsa Miró[1,2], Aina Bellver-Sanchis[3], Christian Griñán-Ferré[3], Mercè Pallàs[3], Anna Pérez-Bosque[2], Miquel Moretó[2], Laura Pont[4,5], Fernando Benavente[5], José Barbosa[5], Carmen Rodríguez[1], Javier Polo[1]

1 APC Europe S.L.U., Granollers, Spain, 2 Departament de Bioquímica i Fisiologia (Secció de Fisiologia), Facultat de Farmàcia i Ciències de l'Alimentació and Institut de Nutrició i Seguretat Alimentària, Universitat de Barcelona (INSA·UB), Barcelona, Spain, 3 Departament de Farmacologia, Toxicologia i Química Terapèutica (Secció de Farmacologia) Facultat de Farmàcia i Ciències de l'Alimentació and Institut de Neurociències (CIBERNED), Universitat de Barcelona, Barcelona, Spain, 4 Department of Chemical Engineering and Analytical Chemistry, Institute for Research on Nutrition and Food Safety (INSA·UB), University of Barcelona, Barcelona, Spain, 5 Serra Húnter Programe, Generalitat de Catalunya, Barcelona, Spain

* marc.sola@apc-europe.com

**Data Availability Statement:** All relevant data are within the manuscript and its Supporting Information files.

## Abstract

In the present study, an enzymatically hydrolyzed porcine plasma (EHPP) was nutritionally and molecularly characterized. EHPP molecular characterization showed, in contrast to spray-dried plasma (SDP), many peptides with relative molecular masses ($M_r$) below 8,000, constituting 73% of the protein relative abundance. IIAPPER, a well-known bioactive peptide with anti-inflammatory and antioxidant properties, was identified. In vivo functionality of EHPP was tested in *C. elegans* and two different mouse models of intestinal inflammation. In *C. elegans* subjected to lipopolysaccharide exposure, EHPP displayed a substantial anti-inflammatory effect, enhancing survival and motility by 40% and 21.5%, respectively. Similarly, in mice challenged with *Staphylococcus aureus* enterotoxin B or *Escherichia coli* O42, EHPP and SDP supplementation (8%) increased body weight and average daily gain while reducing the percentage of regulatory Th lymphocytes. Furthermore, both products mitigated the increase of pro-inflammatory cytokines expression associated with these challenged mouse models. In contrast, some significant differences were observed in markers such as *Il-6* and *Tnf-α*, suggesting that the products may present different action mechanisms. In conclusion, EHPP demonstrated similar beneficial health effects to SDP, potentially attributable to the immunomodulatory and antioxidant activity of its characteristic low $M_r$ bioactive peptides.

## Introduction

Spray-dried plasma (SDP) is a complex protein-rich ingredient obtained from the industrial fractionation of blood from healthy farm animals grown for human consumption. Several

**Funding:** Funding for this study was provided by APC Europe, S.L.U., Granollers, Spain that is a company that manufacture animal blood products for animal consumption. The company provided support in the form of salaries for authors M.S.-G., C.R., and J.P. retrospectively, but the company did not have any additional role. The specific roles of these authors are articulated in the 'author contributions' section. In addition, the authors would like to acknowledge the Centro para el Desarrollo Tecnológico Industrial (CDTI project IDI-20180886) for co-funding this work. The funders had no role in study design, data collection and analysis, decision to publish, or preparation of the manuscript.

**Competing interests:** The authors have declared that no competing interests exist.

studies have reported positive effects of animal feeds containing SDP on performance, morbidity, and mortality. These positive effects have been documented across multiple species including pigs [1], mice [2], ruminants [3], poultry [4], fish [5], and companion animals [6, 7] and are attributed to its well-known anti-inflammatory properties [8–10].

SDP is a complex mixture that contains bioactive peptides (BAP) and proteins, such as immunoglobulins, transferrin, and albumin, as well as growth factors, hormones, and different cytokines [11, 12], which can be responsible for its anti-inflammatory effects. During the past decade, several studies have contributed to a better understanding of the mechanisms related to the beneficial functional properties associated with SDP supplementation in animal feeds. For instance, feed supplemented with SDP prevented mucosa inflammation of the small intestine induced by intraperitoneal administration of *Staphylococcus aureus* enterotoxin B (SEB) [2, 8, 13] and reduced inflammation and colitis severity in mdr1a-/- mice [2]. Besides, the immune modulatory effects of SDP supplementation are not limited to the intestinal mucosa and have similar effects in other mucosal areas. Indeed, supplementation with SDP reduced acute pulmonary inflammation induced by lipopolysaccharide inhalation [14, 15]. Additionally, dietary SDP improved the pregnancy rate in mice under transport stress by attenuating pro-inflammatory cytokine expression [16]. These studies indicated that SDP supplementation modulates the intensity of the inflammatory response in the intestinal mucosa and modifies the immune response in other mucosal areas throughout the common mucosal immune system (mucosa-associated lymphoid tissue, MALT). Moreover, SDP supplementation reduced the pro-inflammatory cytokine expression and stimulated the abundance of regulatory cells, thus restoring the activated/regulatory ratio [2]. Furthermore, the inclusion of SDP in feed for young mice and pigs modified the intestinal microbiota, especially by increasing the population of the *Lactobacillaceae* family and other commensal bacteria with well-known beneficial effects on animal health [17–19].

Generally, BAP are natural compounds of food or part of the protein that are inactive in the precursor molecule. However, they may be activated after hydrolysis and transported to the active site, where they can regulate many body functions [20]. BAP can be released naturally during the gastrointestinal digestion of food, but they can be also produced by microorganisms in the fermentation process or during the food-processing by hydrolysis of food proteins using exogenous proteases extracted from vegetable tissues, animal tissues, and microbial cells [21]. Protein hydrolysates from milk [22], lysozyme [23], red blood cells [24], plasma [21, 25], and marine organisms have been recommended as excellent sources of BAP and are generally regarded as safe and easy to be absorbed by humans and animals [26].

Interestingly, BAP have been defined as specific protein fragments that may present a positive influence on physiologic and metabolic functions or conditions in the body, hence ultimately producing beneficial effects on human or animal health. Some of these effects include antimicrobial, antihypertensive, antioxidative, anticytotoxic, anticancer, anti-inflammatory, immunomodulatory, opioid and mineral transport activities [27]. Therefore, BAP could be partially responsible for the health benefits associated with SDP supplementation. Additionally, the size of these BAP varies from 10 to 150 amino acids with a net charge between -3 and +20 and a hydrophobic content below 60%. Most of them are cationic and short with less than 50 amino acids with molecular weight under 10,000 Da [28]. They have a diverse amino acid sequence, 3D structures, activity, and mechanism of action. Due to their 3D structure and short amino acid sequence, some BAP are heat resistant to very high temperatures and active in a variety of different pH [29].

Enzymatic hydrolyzed porcine plasma (EHPP), which is a product commercially available (APC Europe S.L.U., Granollers, Spain), is a spray-dried protein hydrolysate produced by enzymatic hydrolysis with broad-specificity bacterial endopeptidase under controlled pH,

temperature, time, and specific substrate-to-enzyme ratio. The final product is a light brown micro-granulated and highly soluble powder with a hydrolysis degree >15% that contains low molecular mass proteins, peptides, and an increased amount of free amino acids.

The aim of this study was to characterize the nutritional properties of EHPP, estimate the molecular mass distribution, and identify the proteinaceous components. In addition, three different animal models of inflammation were used to evaluate *in vivo* the functional properties of EHPP compared to conventional SDP under stress conditions. Specifically, the nematode *Caenorhabditis elegans* (*C. elegans*) model and two models of mild intestinal inflammation in mice. *C. elegans* has emerged as a powerful and straightforward tool for advancing our understanding of the mechanisms of intestinal inflammatory disease in humans and animals [30]. The combination of these animal models allows obtaining an accurate and broad insight into EHPP functionality, disclosing the advantages of EHPP over SDP.

## Materials and methods

### Nutritional characterization

For nutritional characterization of EHPP (APC Europe S.L.U., Granollers, Spain) and porcine SDP (APPETEIN GS®, APC Europe S.L.U.), commercial samples were analyzed for dry matter (AOAC method 925.45), crude protein (AOAC method 990.03), ash (AOAC method 942.05), crude fat (AOAC method 954.02), crude fiber (AOAC method 962.09), calcium (ISO 27085:2009), chloride (AOAC 943.01), sodium (ISO 27085:2009), iron (ISO 27085:2009), potassium (ISO 27085:2009), and phosphorus (ISO 27085:2009) [31]. Additionally, amino acids (ISO 13903:2005), including tryptophan (ISO 13904:2016), were also analyzed.

### Molecular characterization

**Size-exclusion chromatography.** Size-exclusion chromatography (SEC) was performed under isocratic elution conditions with a HiLoad 16/600 Superdex 200 preparative grade column (10,000–600,000 relative molecular mass ($M_r$) range, GE Healthcare, Chicago, IL, USA) and a HiLoad 16/600 Superdex 75 preparative grade column (3,000–70,000 $M_r$ range, GE Healthcare) for the analysis of SDP and EHPP, respectively, using an HP 1100 series liquid chromatograph with an ultraviolet absorption diode array detector (Agilent Technologies, Waldbronn, Germany). Instrument control, data acquisition and data processing were performed using the Chemstation LC3D software (Agilent Technologies). The mobile phase consisted of phosphate buffered saline (PBS, pH = 7.50) and absorbance was monitored at 214 nm. After column equilibration with mobile phase (two column volumes), a calibration curve was constructed by injecting 1 mL of a 500 mg/L mixture of protein standards of known $M_r$ according to the manufacturer instructions. SDP and EHPP were analyzed by injecting volumes of 1 mL, and fractions (for EHPP only) were collected using an Agilent 1200 series analytical fraction collector (Agilent Technologies).

**Sample desalting, purification and preconcentration.** Before matrix-assisted laser desorption ionization time-of-flight mass spectrometry (MALDI-TOF-MS) analyses, EHPP fractions (around 10 mL each one) were evaporated until dryness (SpeedVac™ concentrator, Thermo Fisher Scientific, Waltham, MA, USA), redissolved in a small volume of water (1 mL), desalted, and preconcentrated using Oasis® HLB 96-well μElution plates (Waters, Milford, MA, USA) as described in a previous work with minor modifications [32]. Five hundred μL of sample solution were loaded and the 50 μL eluates were evaporated until dryness, before reconstitution in water to a final volume of 5 μL.

Fraction 2' of EHPP was further desalted and purified before performing liquid chromatography-tandem mass spectrometry (LC-MS/MS). The fraction was cleaned through C18 tips

(reversed phase, P200 TopTip C18, PolyLC, Columbia, MD, USA) and strong cation exchange (SCX) columns (Waters) following the manufacturer instructions. The elutes from SCX were evaporated until dryness and reconstituted in 3% (v/v) ACN solution with 1% (v/v) formic acid (FA) to a final volume of 13 μL.

**MALDI-TOF-MS.** MALDI-TOF mass spectra for EHPP fractions were obtained using a 4800 MALDI TOF/TOF mass spectrometer (Applied Biosystems, Waltham, MA, USA) as described in a previous work [33]. Mass spectra were acquired between 600–1,000 m/z using the low mass positive mode, 1,000–25,000 m/z using the linear mid mass positive mode and 25,000–150,000 m/z using the linear high mass positive mode.

**LC-MS/MS.** LC-MS/MS analyses for Fraction 2' of EHPP were performed on an Orbitrap Fusion™ Lumos™ (Thermo Fisher Scientific) coupled to a Dionex Ultimate 3000 chromatographic system (Thermo Fisher Scientific). Six μL of the sample were diluted in 3% (v/v) ACN solution with 1% (v/v) FA and loaded onto a 300 μm × 5 mm, 5 μm, 100 Å, C18 PepMap100 μ-precolumn (Thermo Fisher Scientific) at a flow rate of 15 μL/min. Peptides were separated using a 75 μm × 250 mm, 1.8 μm, 100 Å, C18 NanoEase MZ HSS T3 analytical column (Waters) with a 120 min run, comprising three consecutive steps with linear gradients at 250 nL/min from 3 to 35% B in 90 min, from 35 to 50% B in 5 min, and from 50 to 85% B in 2 min, followed by isocratic elution at 85% B in 5 min and stabilization to initial conditions (A = 0.1% (v/v) FA in water and B = 0.1% (v/v) FA in ACN).

The mass spectrometer was operated in ESI+ using the data-dependent acquisition mode. MS scans were acquired with a resolution of 120,000 in the range comprised between 350 and 1,800 m/z. The most intense ions per scan were fragmented by higher-energy C-trap dissociation (HCD). The ion count target values were 400,000 and 10,000 for MS and MS/MS scans, respectively. Target ions already selected for MS/MS were dynamically excluded for 15s. Spray voltage in the Advion TriVersa NanoMate source (Advion, Ithaca, NY, USA) was set to 1.60 kV and the ion transfer tube temperature was 275˚C. Experimental data were analyzed with Xcalibur (version 4.2.28.14, Thermo Fisher Scientific) and database searches for the identification of peptides and their protein origin were performed with Proteome Discoverer (version 2.1.1.21 software, Thermo Fisher Scientific) using the algorithms ProSightPD cRAWler node with High/High Xtract (against the proteinaceous warehouse database from *Sus Scrofa*), and SEQUEST HT (against the SwissProt database from *Sus Scrofa*), respectively. Search parameters were set to allow for dynamic modifications of methionine oxidation, acetylation on N-terminus, phosphorylation on serine, threonine, and tyrosine, and deamidation on asparagine and glutamine. Peptide mass tolerance was 10 ppm and the MS/MS tolerance 0.02 m/z. Peptides with a false discovery rate (FDR)<1% were considered as positive identifications with a high confidence level.

## Functional evaluation of EHPP in *C. elegans*

*C. elegans* **maintenance and treatment.** The wild-type (WT) *C. elegans* strain Bristol N2, and the transgenic strain CL2006, which expresses human Aβ$_{1–42}$, were used. Standard methods were applied for culturing and observing *C. elegans*. N2 were propagated at 20˚C, while CL2006 worms were maintained at 16˚C in a temperature-controlled incubator, both on a solid nematode growth medium (NGM, 17 g/L bacto-agar, 2.5 g/L bacto-peptone, 3 g/L NaCl, 1 mL/L 1 mM CaCl$_2$, 4 mL/L 25 mM KPO$_4$ (pH = 6.0), 1 mL/L 1 mM MgSO$_4$, and 1 mL/L 5 mg/mL cholesterol) seeded with *E. coli* OP50 strain as a food source. To obtain the age-synchronized population of eggs, gravid adults were treated with an alkaline hypochlorite solution (0.5 M NaOH, ∼2.6% NaCl) for 5–7 min. Fertilized eggs were suspended in S-medium (30 μL 1M MgSO$_4$, 30 μL 1 mM CaCl$_2$, 100 μL 100x Trace metal solution, 100 μL 1M K$_3$C$_6$H$_5$O$_7$

(pH = 6.0), and S-basal to 10 mL; S-basal: 5.85 g/L NaCl, 1 g/L $K_2HPO_4$, 6 g/L $KH_2PO_4$, 8μL/L 5 mg/mL cholesterol, 1000 mL/L dd$H_2O$) for 12 h, and first stage larvae (L1) were allowed to hatch overnight in the absence of food.

Drug assays were performed in 96-well plate format, in liquid culture, and treated for 4 days at 20˚C. Each well contained a final volume of 60 μL, comprising 25–30 worms in the L1 stage diluted in S-medium, EHPP or SDP at the appropriate doses and OP50 inactivated by freeze-thaw cycles and suspended in S-medium complete (30 μL 1M $MgSO_4$, 30 μL 1 mM $CaCl_2$, 100 μL 100x Trace metal solution, 100 μL 1M $KH_2PO_4$ (pH = 6.0), 8 μL 5 mg/mL cholesterol, 50 μL Penicillin/Streptomycin, 50 μL Nystatin, and S-basal to 10 mL) to a final optical density at 595 nm ($OD_{595}$) of 0.9–0.8 measured in the microplate reader.

Adult hermaphrodites of each strain were transferred to a new NGM plate every 4 days to maintain and avoid food deprivation. Nematodes were transferred using a worm picker, which consists of a 1-inch piece of 32-gauge platinum wire plugged into the tip of a Pasteur pipette. The platinum wire heats and cools quickly, helping to prevent contamination of worm stocks.

To prevent SDP and EHPP metabolization effects, inactivated *E. coli* OP50 was used in all assays as a food source. OP50 bacteria were cultured in LB media (Sigma-Aldrich, St. Louis, MI, USA) for 16 h, 37˚C, 200 rpm in a heating thermoshaker. After growth, bacteria suspension was distributed in tubes and centrifuged at 4˚C and 4,000 *g* for 30 min. Supernatants were discarded and the pellets inactivated by 3 cycles of freeze/thaw, alternating liquid nitrogen and water bath at 37˚C. Finally, the pellets of OP50 were frozen one last time using liquid nitrogen and stored at –80˚C.

**Food clearance assay.** WT worms were subjected to a drug assay as described above, except for the OP50 bacteria $OD_{595}$ adjusted to 0.7. Worms were grown with continuous shaking for 6 days at 180 rpm and 20˚C. Tested concentrations ranged between 1 mg/mL and 10 mg/mL. For control wells, 1% dimethyl sulfoxide (DMSO, vehicle) and 5% DMSO (toxic condition) were used; and for blank wells S-medium and S-medium complete only, without eggs or OP50, were respectively used. The effect of compounds on *C. elegans* physiology was monitored by the rate at which the OP50 suspension was consumed, as a readout for *C. elegans* growth, survival, or fecundity. The $OD_{595}$ was measured in triplicate samples daily.

**Oxidative tolerance assay.** To investigate sensitivity to oxidative stress after EHPP or SDP supplementation, N2 adults were transferred onto plates that included 6.2 mM t-butyl hydroperoxide (Sigma-Aldrich) in nematode growth medium agar. Worms were incubated on these plates at 20˚C and scored as dead when they did not respond to repeated prodding with the worm picker.

**Motility assay.** N2 (WT) strain and the transgenic strain CL2006, which presents age-dependent paralysis, were used to evaluate the impact of experimental feeds on motor dysfunction. At day 4 of age, worms were transferred from the 96-well plates onto an unseeded NGM plate. Plates were allowed to dry for 45 min before starting the motility assays. Motility assays were performed at 20˚C in 30 mm NGM plates and the whole surface of the plate was covered by OP50. Five to 10 adult nematodes were placed in the center of a circle (with 1 cm of diameter), in each unseeded 30 mm NGM plates (a minimum of 50 worms were scored in each replicate). After 1 min, the number of worms remaining inside the circle were scored as the percentage of locomotor defective. The motility assay was run in triplicates (n = 3), with a total of at least 150 worms tested per compound concentration.

**Lipopolysaccharide assay.** L4 (3 days old) synchronized worms from NGM plates were washed 3 times with M9 buffer (3 g/L $KH_2PO_4$, 6 g/L $Na_2HPO_4$, 5 g/L NaCl, 1 mL/L 1 mM $MgSO_4$, 800 mL/L dd$H_2O$). Treatment of *C. elegans* with SDP or vehicle was conducted in liquid media (S-media) without OP50 in a 96-well plate. *C. elegans* were treated with EHPP, SDP

or the vehicle at 30 min after lipopolysaccharide (LPS) injury, which was induced by exposing the worms to 100 μg/mL of LPS at 20˚C for 24 h, under constant shaking for oxygenation. After the 24-hour exposure, worms were washed with M9 buffer 3 times and transferred to foodless NGM plates. Before behavioral assays, worms were left for 30 min to acclimate to the plates. Reversal, omega turn, and pause were utilized to evaluate the behaviors of worms in 15-second intervals. Any backward movement of the entire body was scored as a reversal. Omega turns were visually identified by the head nearly touching the tail, as this shape resembled the Greek letter Omega. Worms were scored by an investigator blinded to chemical compound treatments.

## Functional evaluation of EHPP in two mice models of intestinal inflammation

**Animal and feeds.** Procedures used in the study were approved by the Ethic Committee for Animal Experimentation (CEEA) from University of Barcelona and Generalitat of Catalonia (*E. coli* infection: references 290/19 and 10969, respectively; and for SEB inflammation: references 87/17 and 9515, respectively) and followed the recommendations from Federation of European Laboratory Animal Science Associations [34]. C57BL/6 mice (Envigo, Bresso, Italy) were kept under stable temperature and humidity conditions, with a 12 h light–12 h dark cycle and free access to food and water. Mice were weaned at 21 days old and fed the experimental feeds for two weeks (Table 1). The mice were monitored for food intake and body weight throughout the experimental period (until 35 days old).

For both models, there was a non-challenged group fed with control feed (CTL), and three challenged groups: CTL: Control mice fed the control feed; SDP: mice fed with 8% SDP feed; EHPP: mice fed with 8% EHPP feed.

**Table 1. Composition of experimental feeds used in mice.**

| Ingredients | Control feed | SDP[a] feed | EHPP[a] feed |
|---|---|---|---|
| | | g/kg | |
| SDP | - | 80 | - |
| EHPP | - | - | 80 |
| Dried skim milk | 530.7 | 370.1 | 370.1 |
| Corn starch | 199.3 | 335.7 | 335.7 |
| Sucrose | 94.5 | 102.7 | 102.7 |
| Soybean oil | 70.0 | 76.1 | 76.1 |
| Cellulose | 50.0 | 54.3 | 54.3 |
| AIN-93-G-MX (94046)[b] | 35.0 | 38.0 | 38.0 |
| AIN-93 VX (94047)[b] | 15.0 | 16.3 | 16.3 |
| Choline bitartrate | 3.0 | 3.3 | 3.3 |
| Methionine | 2.5 | 3.5 | 3.5 |
| **Nutritional composition** | | | |
| Energy (Kcal/g) | 3.7 | 3.7 | 3.7 |
| Protein | 186 | 180 | 180 |
| Carbohydrates | 567 | 569 | 569 |
| Fat | 73 | 74 | 74 |
| Lysine | 14.9 | 14.9 | 14.5 |

[a]SDP (spray-dried plasma, commercial name: APPETEIN GS®) and EHPP (enzyme hydrolyzed porcine plasma) were provided by APC Europe S.L.U., Granollers, Spain.
[b]AIN-93 VX vitamin mix and AIN-93-G-MX mineral mix were provided by Envigo, Bresso, Italy.

**Intestinal inflammation induced by SEB.** The procedure was performed as previously described [2]. Briefly, on day 34, challenged mice were intraperitoneally administered with the *S. aureus* enterotoxin B (SEB; 25 μg/mice; Toxin Technology, Sarasota, FL, USA), while non-challenged mice received vehicle. Twenty-four hours later, mice were anesthetized by intraperitoneal administration of ketamine:xylazine (100:10 mg/kg) and mesenteric lymph nodes (MLN) and jejunum mucosa were obtained and processed.

**Intestinal infection with *E. coli*.** To induce the intestinal infection, cimetidine (50 mg/kg; Sigma-Aldrich, St. Louis, MI, USA) was administrated intraperitoneally on day 30 of age and, 3 h later, *E. coli* strain 042 was inoculated by oral gavage as previously described [35]. Non-challenged mice received vehicle. Five days after *E. coli* administration (35 days old), mice were anesthetized by intraperitoneal administration of ketamine:xylazine (100:10 mg/kg) and MLN were obtained.

**Mesenteric lymph nodes isolation.** Leukocytes from MLN were obtained as previously described [2]. Briefly, the MLN were incubated in the solution of digestion composed by RPMI-1640 (Gibco, Carlsbad, CA, USA) with 5% inactivated fetal bovine serum (FBS), 100,000 U/L penicillin, 100 mg/L streptomycin, 10 mM HEPES, 2 nM L-glutamine and 150 U/mL collagenase (Gibco) at 37˚C in a thermoshaker. The MLN were mechanically disaggregated, and the cell suspension was centrifuged at 500 *g* for 10 min at 4˚C. The pelleted cells were resuspended in PBS-FBS.

**Cell staining.** Briefly, staining was done on $3 \cdot 10^5$ cells, following the protocol described previously [36]. Cells were analyzed in the Gallios Flow cytometer (Beckman Coulter, Miami, FL, USA) located in the Cytometry Unit of the Technical Services of the University of Barcelona at the Barcelona Science Park. Results were analyzed using the Flowjo Software (Treestar Inc., Ashland, OR, USA).

**Real-time PCR.** Total RNA of jejunum mucosa or leukocytes from MLN was extracted with TRIzol™ reagent (Life Technologies, Carlsbad, CA, USA) following the instructions of the manufacturer. RNA extraction of samples and reverse transcription were done as described previously [37]. The primers used are indicated in Table 2. Product fidelity was confirmed by melt curve analysis. Quantification of the target gene transcripts was performed using hypoxanthine phosphoribosyltransferase 1 (*Hprt1*) as a reference gene and was done using the $2^{-\Delta\Delta CT}$ method [38].

## Statistical analysis in functional evaluation

All statistical analyses were conducted using GraphPad Prism 9 (GraphPad Software Inc., La Jolla, CA, USA). Statistical differences were considered significant at $p < 0.05$. Experiments with *C. elegans* groups were compared with a one-way analysis of variance (ANOVA), followed by Tukey's post hoc test. In some cases, comparison between groups was also performed by two-tailed Student's t-test for independent samples. The statistical outliers were determined with Grubs' test and removed from the analysis. Assays were run in triplicates (n = 3), with a total of at least 150 worms.

**Table 2. Primers used for real-time PCR.**

| Primer[a] | Forward (5′-3′) | Reverse (5′-3′) | Fragment size (bp) |
|---|---|---|---|
| *Il-1*β | GGT CAA AGG TTT GGA AGC AG | TGT GAA ATG CCA CCT TTT GA | 94 |
| *Il-6* | TGT GAA ATG CCA CCT TTT GA | GGT CAA AGG TTT GGA AGC AG | 109 |
| *Hprt1* | TGG ATA CAG GCC AGA CTT TGT T | CAG ATT CAA CTT GCG CTC ATC | 163 |

[a]*Il-1*β, interleukin 1β; *Il-6*, interleukin 6; *Hprt1*, hypoxanthine phosphoribosyltransferase 1

In experiments to induce intestinal inflammation with SEB or *E. coli*, Grubb's test and Shapiro-Wilk's test were performed to detect outlier values and to verify data normality, respectively. To compare groups, one-way ANOVA followed by the Fisher's post hoc test was used for normally distributed data; otherwise, the non-parametric Kruskal-Wallis test was used. When normally distributed data showed no sphericity, an ANOVA with Brown-Forsythe and Welch's test was applied. To analyze the evolution of body weight, the area under the curve was calculated for each animal and these values were compared using the one-way ANOVA.

## Results

### Nutritional composition

EHPP showed a similar amino acid concentration profile to SDP (Table 3). Since it is an hydrolyzed product, its solubility is higher than SDP (97.0% vs 88.0%). As expected, due to the addition of chemicals to maintain the pH during the enzymatic hydrolysis, ash content was higher than in SDP (15.3% vs 9.6%) and, consequently, crude protein was slightly lower (74.6% vs 80.5%). Crude fat was similar for SDP and EHPP (2.6% and 2.4%, respectively) and crude fiber was very low for both products. Minerals including iron, chloride, sodium, phosphorous, and potassium were higher in the EHPP product, which is in concordance with the increased ash content (Table 3).

### Molecular characterization

**SEC.** SEC analyses were performed to estimate the $M_r$ of the components present in EHPP, as well as to prove the hydrolysis effectiveness by comparing the $M_r$ profile of EHPP with that obtained for SDP. First, SEC calibration was performed by analyzing a mixture of protein standards of known $M_r$, which were selected according to the $M_r$ working range of the SEC column. Two SEC columns with different working $M_r$ ranges were used, according to the expected differences in the $M_r$ profiles for SDP and EHPP samples. Fig 1 shows (i) the chromatograms obtained by SEC (at 214 nm) for (A) SDP and (B) EHPP and (ii) the relative abundance for the detected fractions. As expected, the chromatographic profiles for SDP and EHPP were different, suggesting changes in the $M_r$ profile after enzymatic hydrolysis. The $M_r$ of the different fractions were estimated considering the previous $M_r$ calibration (for both columns, calibration curves were obtained with $R^2 > 0.99$). As shown in Fig 1A-i, three fractions were observed for SDP. Fraction 1 contained a heterogeneous mixture of proteins with $M_r$ comprised between 350,000–1,000,000 (26% abundance, see Fig 1A-ii), Fraction 2 (31% abundance) contained a mixture of proteins with $M_r$ between 100,000–350,000 (the peak maximum in Fraction 2 with a $M_r$ of around 150,000 probably corresponded to immunoglobulins), and Fraction 3 (43% abundance) contained a mixture of proteins with $M_r$ comprised between 45,000–100,000 (the peak maximum in Fraction 3 with a $M_r$ of around 66,000 probably corresponded to serum albumin). Compared to SDP, EHPP did not contain proteins in the $M_r$ comprised between 100,000–1,000,000, hence confirming the hydrolysis effectiveness. As can be seen in Fig 1B, Fraction 1' of EHPP contained a heterogeneous mixture of proteins with $M_r$ comprised between 45,000–100,000 (27% abundance, Fig 1B-ii), Fraction 2' contained a mixture of peptides with $M_r$ between 1,000–8,000 (69% abundance) and Fraction 3' contained a mixture of peptides with $M_r$ comprised between 600 and 1,000 (4% abundance).

**MALDI-TOF-MS.** To improve the reliability of the $M_r$ estimation for EHPP, all the SEC fractions were collected and subjected to MALDI-TOF-MS analysis. Fig 2 shows the mass spectra acquired over a range of 25,000–150,000 m/z for Fraction 1'(A), 1,000–25,000 m/z for Fraction 2'(B) and 600–1,000 m/z for Fraction 3' (C) (note that only m/z ranges where peaks were detected are shown). As can be observed in Fig 2A, the mass spectrum for Fraction 1'

**Table 3. Nutritional composition of the enzymatic hydrolyzed porcine plasma (EHPP, APC Europe S.L.U., Granollers, Spain) and spray-dried plasma (SDP, commercial name: APPETEIN GS®, APC Europe S.L.U.).**

| Component | EHPP | SDP |
|---|---|---|
| Crude protein (% m/m) | 74.6 | 80.5 |
| Dry matter (% m/m) | 97.1 | 93.7 |
| Ash (% m/m) | 15.3 | 9.6 |
| Crude fat (% m/m) | 2.6 | 2.4 |
| Solubility (% m/v) | 97.0 | 88.0 |
| Crude fiber (% m/m) | <0.1 | <0.3 |
| Digestible energy [1] (kcal/kg) | 3887 | 4108 |
| Metabolizable energy [2] (kcal/kg) | 3691 | 3906 |
| **Minerals** | | |
| Iron (mg/kg) | 80 | 60 |
| Calcium (% m/m) | 0.2 | 0.2 |
| Chloride (% m/m) | 1.7 | 0.7 |
| Sodium (% m/m) | 2.7 | 2.2 |
| Phosphorous (% m/m) | 1.7 | 1.3 |
| Potassium (% m/m) | 5 | 0.3 |
| **Amino acids** (% m/m) | | |
| Alanine | 4.0 | 4.2 |
| Arginine | 4.8 | 4.7 |
| Aspartic acid | 8.1 | 7.9 |
| Cystine | 1.6 | 2.8 |
| Glutamic acid | 12.0 | 11.7 |
| Glycine | 2.6 | 3.0 |
| Histidine | 2.5 | 2.8 |
| Isoleucine | 2.9 | 2.9 |
| Leucine | 7.5 | 7.8 |
| Lysine | 6.1 | 6.8 |
| Methionine | 0.7 | 0.7 |
| Phenylalanine | 4.3 | 4.6 |
| Proline | 5.0 | 4.5 |
| Serine | 4.4 | 4.7 |
| Threonine | 4.7 | 4.8 |
| Tryptophan | 1.3 | 1.4 |
| Tyrosine | 3.9 | 3.6 |
| Valine | 5.7 | 5.3 |

[1,2] Both parameters were calculated as described by Ewan [39].

demonstrated the presence of single-charged ions corresponding to proteins with $M_r$ around 47,000, 57,000 and 67,000 (within the $M_r$ range estimated by SEC, 45,000–100,000, Fig 1B). Similarly, as shown in Fig 2B, the mass spectrum for Fraction 2' demonstrated the presence of single-charged ions corresponding to peptides with $M_r$ around 1,000–2,000 and 4,000–5,000 (again, within the SEC $M_r$ range, 1,000–8,000, Fig 1B). Finally, the mass spectrum for Fraction 3' (Fig 2C) demonstrated the presence of single-charged ions from peptides with $M_r$ values lower than 1,000 (SEC $M_r$ range was 600–1,000, Fig 1B). MALDI-TOF-MS confirmed the hydrolysis effectiveness through a rapid and simple characterization of the different EHPP fractions, but LC-MS/MS with a high-sensitivity and accurate mass high-resolution mass spectrometer was necessary for a more detailed identification of the different components. This

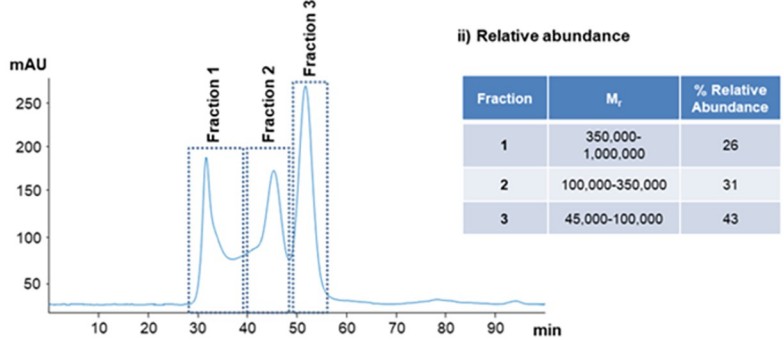

**Fig 1.** (i) Chromatograms obtained by SEC (monitored at 214 nm) and (ii) relative abundances of the detected fractions for (A) SDP and (B) EHPP. A HiLoad 16/600 Superdex 200 preparative grade column (10,000–600,000 $M_r$ range, GE Healthcare) and a HiLoad 16/600 Superdex 75 preparative grade column (3,000–70,000 $M_r$ range, GE Healthcare) were used for the analysis of SDP and EHPP, respectively. Relative abundances were calculated as a percentage, dividing the area of each fraction by the sum of all the fraction areas. EHPP: Enzymatic hydrolyzed porcine plasma; SDP: Spray-dried plasma.

was especially interesting for peptides present in Fraction 2' (the most abundant fraction in EHPP, Fig 1B), which was expected to be the most relevant fraction to explain the bioactivity of the hydrolyzed product.

**LC-MS/MS.** LC-MS/MS analyses were performed to improve the identification of the peptides present in Fraction 2' of EHPP, as well as to correlate them with their protein origin. Table 4 shows that a total number of 92 peptides were detected (with $M_r$ ranging from 750 to 5,500, in agreement with the values estimated by SEC and observed by MALDI-TOF-MS). These peptides corresponded to 39 proteins that were identified with high confidence. As can be seen in this table, the identified proteins are ordered by the "Sum PEP Score". This parameter gives an idea of the identification confidence based on the number of detected peptides for a certain protein, which is correlated, depending on the length of the peptides, with the total coverage of the protein sequence. As an example, actin gamma 1 (accession number A0A0B8RTA2), histone H4 (accession number P62802) and hemoglobin subunit beta (accession number F1RII7) show the three highest "Sum PEP Scores" with 17, 6 and 4 identified

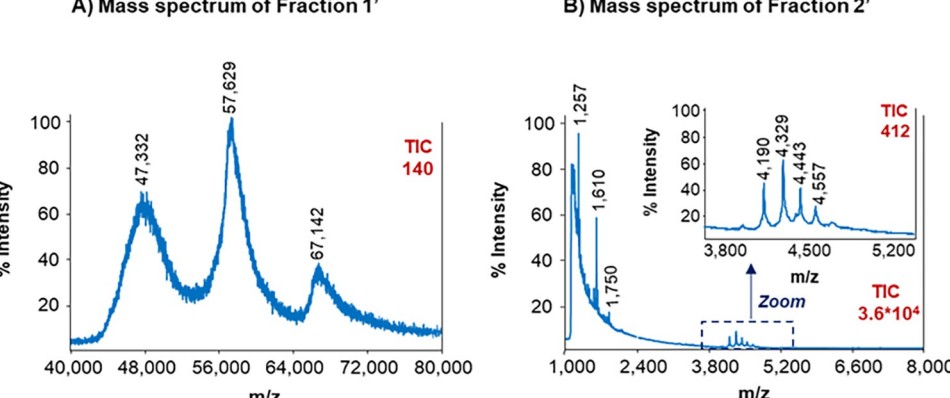

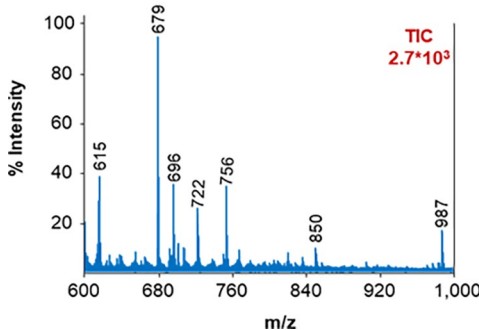

**Fig 2.** MALDI-TOF mass spectra for the collected fractions of EHPP: (A) Fraction 1' (measurement range 25,000–150,000 m/z in the high mass positive mode), (B) Fraction 2' (measurement range 1,000–25,000 m/z in the mid mass positive mode) and (C) Fraction 3' (measurement range 600–1,000 m/z in the low mass positive mode). Mass spectra are shown in the m/z ranges where peaks were detected. EHPP: Enzymatic hydrolyzed porcine plasma.

peptides (corresponding to 37, 25 and 10% of the protein sequence coverage, respectively). Table 4 also shows the biological functions of the identified proteins, which could be relevant to explain the bioactivity of EHPP.

## Functional evaluation of EHPP in *C. elegans*

**Chronic toxicity.**  Fig 3 shows a plot of the OD of the *E. coli* OP50 suspension over the 6 days of the food clearance assay. DMSO 1% corresponded to drug vehicle and was used as a negative (safe) control, whereas DMSO 5% was used as a positive (toxic control). The three different concentrations of SDP and EHPP tested (1, 5 and 10 mg/mL) were classified as safe, as the OD decrease paralleled the plot of vehicle control samples and visual inspection confirmed normal growth of the worms (S1 Table). Then, the EHPP and SDP concentration selected to perform the functional experiments was 5 mg/mL.

**Oxidative stress.**  The effect of EHPP and SDP on oxidative stress was evaluated after a treatment of N2 (WT) strain with tert-butyl (6.2 mM). As can be observed in Fig 4, SDP and EHPP increased protection against oxidative stress in comparison with the vehicle group, improving worm survival by 28% and 40%, respectively (S2 Table). These survival improvements were close to the values for the positive control group with vitamin C (57%).

**Table 4. Protein origin (with the accession number, Sum PEP Score, number of identified peptides, sequence coverage and biological functions) for the 92 peptides identified in Fraction 2' of EHPP by LC-MS/MS.**

| Plasma protein | Accession number | Sum PEP Score | [#]Peptides | Coverage | Biological function[a] |
|---|---|---|---|---|---|
| Actin, gamma 1 | A0A0B8RTA2 | 44 | 17 | 37 | Adherens junction assembly, apical protein localization, axonogenesis, cell motility, cellular response to cytochalasin B, establishment or maintenance of cell polarity, morphogenesis of a polarized epithelium, negative regulation of protein binding, regulation of protein localization to plasma membrane, regulation of transepithelial transport, regulation of transmembrane transporter activity, synaptic vesicle endocytosis |
| Histone H4 | P62802 | 11 | 6 | 25 | DNA-templated transcription initiation, nucleosome assembly |
| Hemoglobin subunit beta | F1RII7 | 10 | 4 | 10 | Cellular oxidant detoxification, hydrogen peroxide catabolic process |
| Tubulin beta chain | F1S6M7 | 9 | 3 | 8 | Axon guidance, microtubule cytoskeleton organization, mitotic cell cycle |
| Uncharacterized protein | F2Z571 | 8 | 3 | 8 | - |
| Histone H2A[b] | I3L7T6 | 6 | 3 | 18 | DNA damage checkpoint signaling, double-strand break repair via homologous recombination, spermatogenesis |
| Tubulin alpha | B6A7R0 | 5 | 3 | 9 | Microtubule-based process |
| Histone H2B[b] | I3LEB6 | 5 | 2 | 19 | Antibacterial humoral response, antimicrobial humoral immune response mediated by antimicrobial peptide, innate immune response in mucosa, negative regulation of tumor necrosis factor-mediated signaling pathway, nucleosome assembly, defense response to Gram-negative bacterium, defense response to Gram-positive bacterium, killing of cells of another organism |
| Albumin | A2THZ2 | 5 | 4 | 4 | Cellular response to starvation, maintenance of mitochondrion location, negative regulation of apoptotic process |
| Cofilin-1 (fragment) | K7GRM2 | 5 | 1 | 22 | Actin filament depolymerization, actin filament fragmentation, cytoskeleton organization, positive regulation of embryonic development, regulation of cell morphogenesis, actin filament organization, actin filament severing, cell motility |
| Tousled like kinase 2 | F1RRU6 | 5 | 4 | 5 | Chromosome segregation, intracellular signal transduction, peptidyl-serine phosphorylation, regulation of chromatin assembly or disassembly, cellular response to DNA damage stimulus, cellular response to gamma radiation, negative regulation of proteasomal ubiquitin-dependent protein catabolic process |
| Mediator of RNA polymerase II transcription subunit 15 | F1RK85 | 5 | 3 | 4 | Regulation of transcription by RNA polymerase II |
| Haptoglobin | B3GQZ2 | 5 | 2 | 4 | Acute-phase response, defense response to bacterium |
| Histone H1.2-like protein | Q4TTS4 | 4 | 2 | 14 | Nucleosome assembly |
| Fibrinogen beta chain | F1RX37 | 4 | 2 | 3 | Blood coagulation, innate and immune response, cell-matrix adhesion, fibrinolysis, induction of bacterial agglutination, negative regulation of endothelial cell apoptotic process, negative regulation of extrinsic apoptotic signaling pathway via death domain receptors, plasminogen activation, platelet aggregation, positive regulation of ERK1 and ERK2 cascade, positive regulation of exocytosis, positive regulation of heterotypic cell-cell adhesion, positive regulation of peptide hormone secretion, positive regulation of vasoconstriction, protein polymerization, response to calcium ion |
| Histone H2B[b] | I3LAQ1 | 4 | 2 | 19 | Antibacterial humoral response, antimicrobial humoral immune response mediated by antimicrobial peptide, innate immune response in mucosa, negative regulation of tumor necrosis factor-mediated signaling pathway, nucleosome assembly |
| Uncharacterized protein | F1S394 | 4 | 2 | 3 | - |
| Transferrin receptor protein 1 | D7RK08 | 4 | 2 | 3 | Cellular iron ion homeostasis, receptor internalization, transferrin transport |
| Uncharacterized protein | I3LIY7 | 4 | 1 | 30 | - |
| Actinin, alpha 4 | A0A0B8S0C5 | 4 | 2 | 2 | Protein transport, regulation of apoptotic process, positive regulation of sodium: proton antiporter activity, tumor necrosis factor-mediated signaling pathway, regulation of nucleic acid-templated transcription |
| Histone H2A[b] | F1RPL3 | 3 | 3 | 14 | DNA damage checkpoint signaling, double-strand break repair via homologous recombination, spermatogenesis |

(*Continued*)

**Table 4.** (Continued)

| Plasma protein | Accession number | Sum PEP Score | #Peptides | Coverage | Biological function[a] |
|---|---|---|---|---|---|
| Trypsinogen | C5IWV5 | 3 | 2 | 7 | Proteolysis |
| Histidine rich glycoprotein | F1SFI5 | 3 | 1 | 2 | Antimicrobial humoral immune response mediated by antimicrobial peptide, cytolysis by host of symbiont cells, heme transport, defense response to fungus, negative regulation of angiogenesis, negative regulation of cell adhesion mediated by integrin, negative regulation of cell growth, negative regulation of endopeptidase activity, negative regulation of fibrinolysis, negative regulation of vascular endothelial growth factor signaling pathway, platelet activation, positive regulation of apoptotic process, positive regulation of blood vessel remodeling, positive regulation of immune response to tumor cell, regulation of gene expression, regulation of peptidyl-tyrosine phosphorylation, regulation of platelet activation, regulation of protein-containing complex assembly |
| Fibrinogen A-alpha-chain | Q28936 | 3 | 1 | 3 | Blood coagulation |
| Neurofilament medium polypeptide | F1RJU6 | 3 | 2 | 2 | Neurofilament bundle assembly |
| Myelin basic protein | I3LMX2 | 3 | 1 | 6 | Myelination |
| Pyruvate kinase | F1SHL9 | 2 | 1 | 2 | Glycolysis, ATP biosynthetic process |
| Cofilin-1 | P10668 | 2 | 1 | 11 | Actin filament depolymerization, actin filament fragmentation, cytoskeleton organization, positive regulation of embryonic development, regulation of cell morphogenesis |
| Zyxin | F1SRV9 | 2 | 2 | 6 | Cell-matrix adhesion, cellular response to interferon-gamma, integrin-mediated signaling pathway, signal transduction, stress fiber assembly, transforming growth factor beta receptor signaling pathway |
| Thymosin beta 4 X-linked | B7TJ02 | 2 | 1 | 55 | Actin filament organization |
| Uncharacterized protein | I3LB54 | 2 | 1 | 1 | - |
| GTP-binding nuclear protein Ran | F1RFQ7 | 2 | 1 | 13 | Protein import into nucleus, ribosomal subunit export from nucleus |
| Heat shock 70 kDa protein 1-like | A5A8V7 | 2 | 1 | 2 | Cellular response to unfolded protein, binding of sperm to zona pellucida, chaperone cofactor-dependent protein refolding, positive regulation of protein targeting to mitochondrion, vesicle-mediated transport, protein refolding |
| Inter-alpha-trypsin inhibitor heavy chain H2 | O02668 | 1 | 1 | 1 | Hyaluronan metabolic process |
| E3 ubiquitin-protein ligase UBR5 | K9IVL0 | 1 | 1 | 0 | Cellular response to DNA damage stimulus, negative regulation of histone H2A K63-linked ubiquitination, protein polyubiquitination, regulation of double-strand break repair |
| Alpha-2-macroglobulin | K9J6H8 | 1 | 1 | 1 | Negative regulation of complement activation, lectin pathway, stem cell differentiation |
| Uncharacterized protein | F1S6C8 | 1 | 1 | 2 | - |
| Adenylyl cyclase-associated protein | A0A0B8S0B1 | 1 | 1 | 3 | Actin cytoskeleton organization, cell morphogenesis |
| Actin related protein 2/3 complex subunit 2 | B5APU7 | 1 | 1 | 6 | Actin filament polymerization, Arp2/3 complex-mediated actin nucleation |

[a]Biological functions for the 39 pig plasma proteins were searched against the UniprotKB database (www.uniprot.org).

[b]Despite Histone H2A and Histone H2B appear twice in the table, they correspond to different entries from the database.

EHPP: Enzymatic hydrolyzed porcine plasma.

**Motility.** A motility assay was performed to assess motor function, where worms were placed in the center of a plate, in the presence of food, and allowed to explore the environment freely for 1 minute. N2 and CL2006 strains were treated with DMSO 1% as a drug vehicle, which should not cause a relevant effect on locomotor phenotype. As expected with drug vehicle, the percentage of WT worms that stayed in the circle after 1 min was the lowest (~16%),

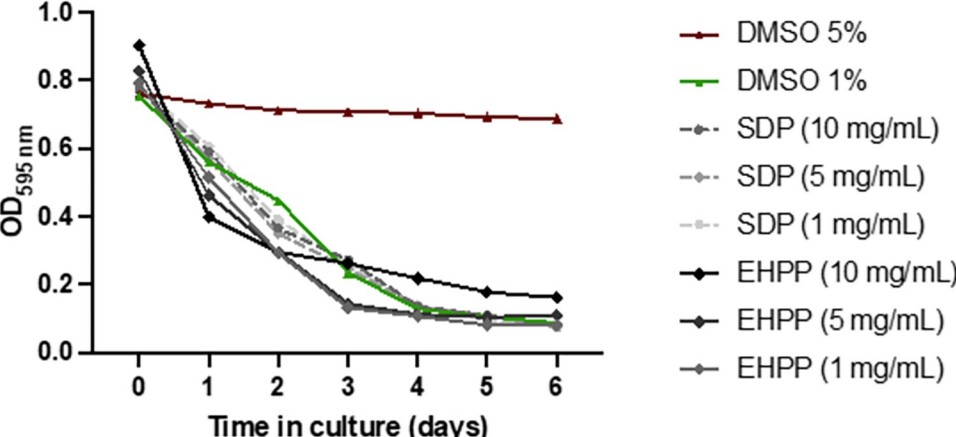

**Fig 3. EHPP and SDP toxicity were evaluated using the food clearance assay.** Statistical analysis: Application of a non-linear regression model for a sigmoidal curve against DMSO 1% presented a unique model for LogIC50 and Hill Slope values, suggestive of no statistical differences between curves (DMSO 5% vs all groups: p<0.0001). EHPP: Enzymatic hydrolyzed porcine plasma; SDP: Spray-dried plasma; DMSO: dimethyl sulfoxide; OD: optical density.

whereas CL2006 worms showed a significant percentage of motility defects (~60%) due to age-dependent paralysis (p<0.001, Fig 5). Strikingly, after the chronic treatment of CL2006 worms with EHPP and SDP at 5 mg/mL, a significant improvement of 21.5% and 35%, respectively,

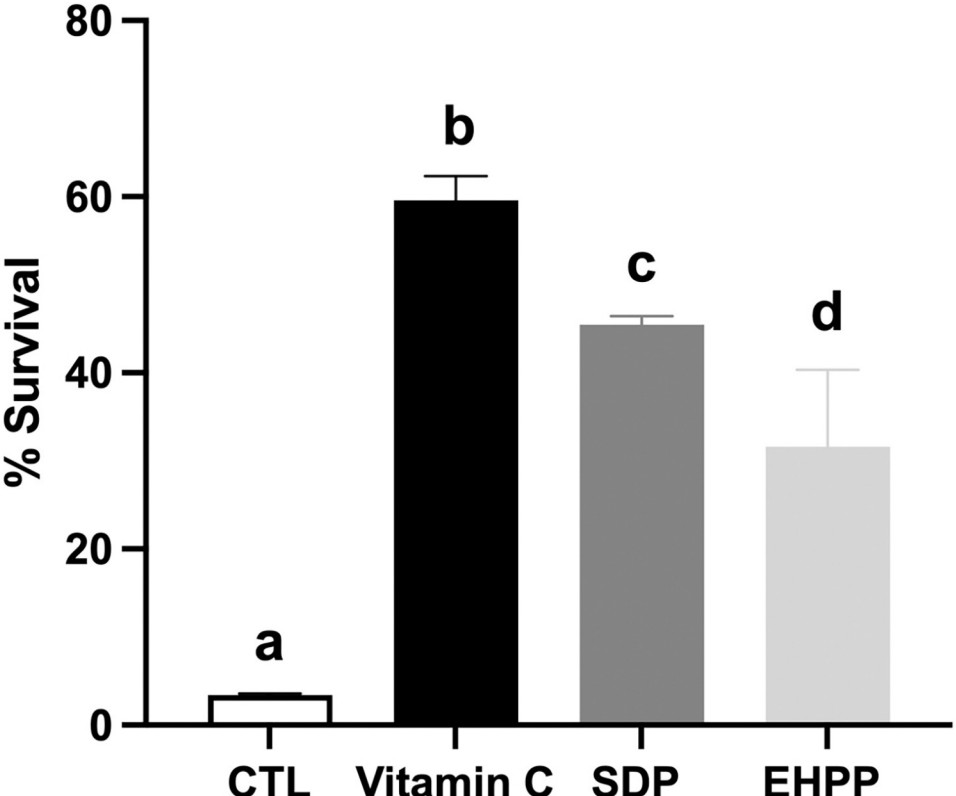

**Fig 4. Summary of oxidative stress response of different *C. elegans* groups (N2, WT) after treatment with EHPP (5 mg/mL) and SDP (5 mg/mL) supplementation or Vitamin C (58 μM).** Statistics: one-way ANOVA (Tukey test). Results are expressed as means ± SEM (n = 3, 100 worms in each group/replicate). Means with different letters are significantly different (p<0.05). CTL: control; EHPP: Enzymatic hydrolyzed porcine plasma; SDP: Spray-dried plasma.

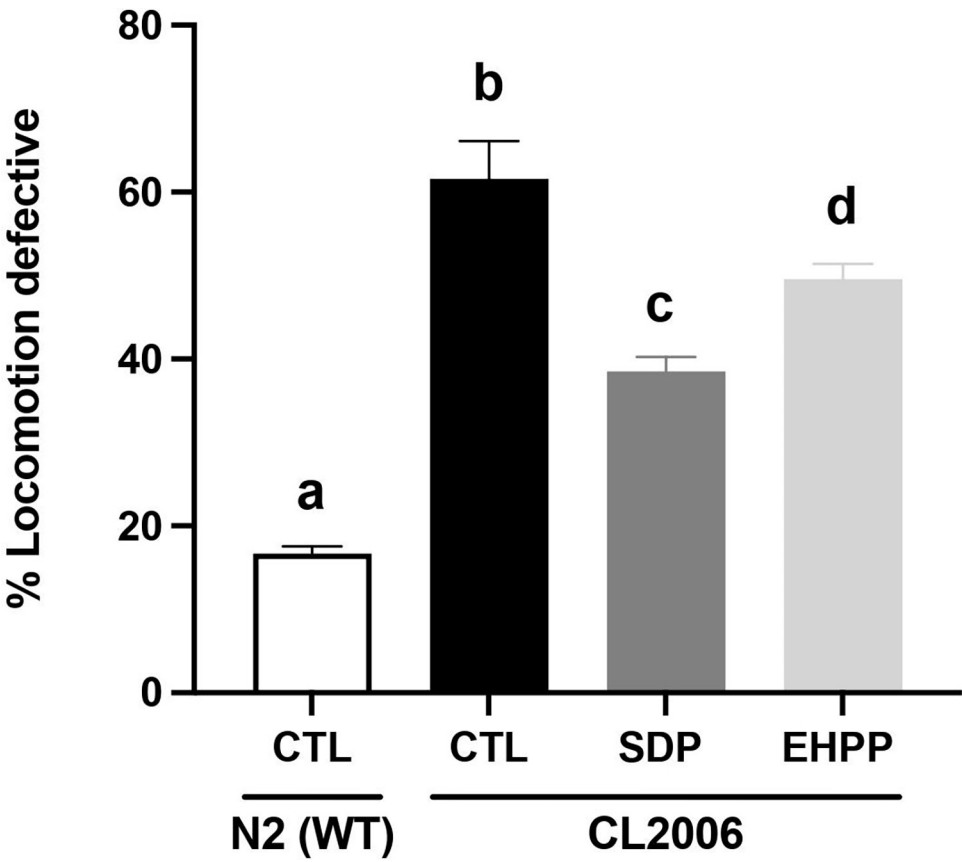

**Fig 5. Effects of EHPP (5 mg/mL) and SDP (5 mg/mL) treatments on age-dependent paralysis in CL2006 worms.**
Statistics: one-way ANOVA (Tukey test). Results are expressed as means ± SEM (n = 3, 50 worms in each group/
replicate). Means with different letters are significantly different (p<0.05). CTL: control; EHPP: Enzymatic hydrolyzed
porcine plasma; SDP: Spray-dried plasma.

in the defective motility behavior was found (Fig 5 and S3 Table). Therefore, both treatments
ameliorated age-dependent paralysis to a significant extent.

**LPS-induced injury.** To evaluate the mechanism of innate immunity and stress response,
an LPS-induced injury assay was performed, since LPS modulates the expression of selected
host immune- and aging-related genes in *C. elegans*. Endpoints of reversal, omega turn, and
pause were chosen to evaluate the locomotion behavioral phenotypes after LPS exposure.
These criteria were selected because worms rarely make dorsal turns. As can be observed in
Fig 6, LPS exposure resulted in a significant increase in the frequencies of altered phenotypes,
which effects were reversed by SDP and EHPP. Interestingly, it was found a higher number of
reversals in the EHPP treated group compared to the LPS 24 h treated group, whereas no sig-
nificant changes were found in SDP treated group (Fig 6A). Furthermore, a reduction in num-
ber of omega turn was only found in EHPP treated group (p<0.01, Fig 6B). Ultimately, a
significantly decreased number of pause behaviors in both treated groups in comparison with
the LPS 24 h treated group (Fig 6C) was shown (S4 Table). Collectively, those results demon-
strated the highest anti-inflammatory effects of EHPP against LPS-induced behavioral
phenotypes.

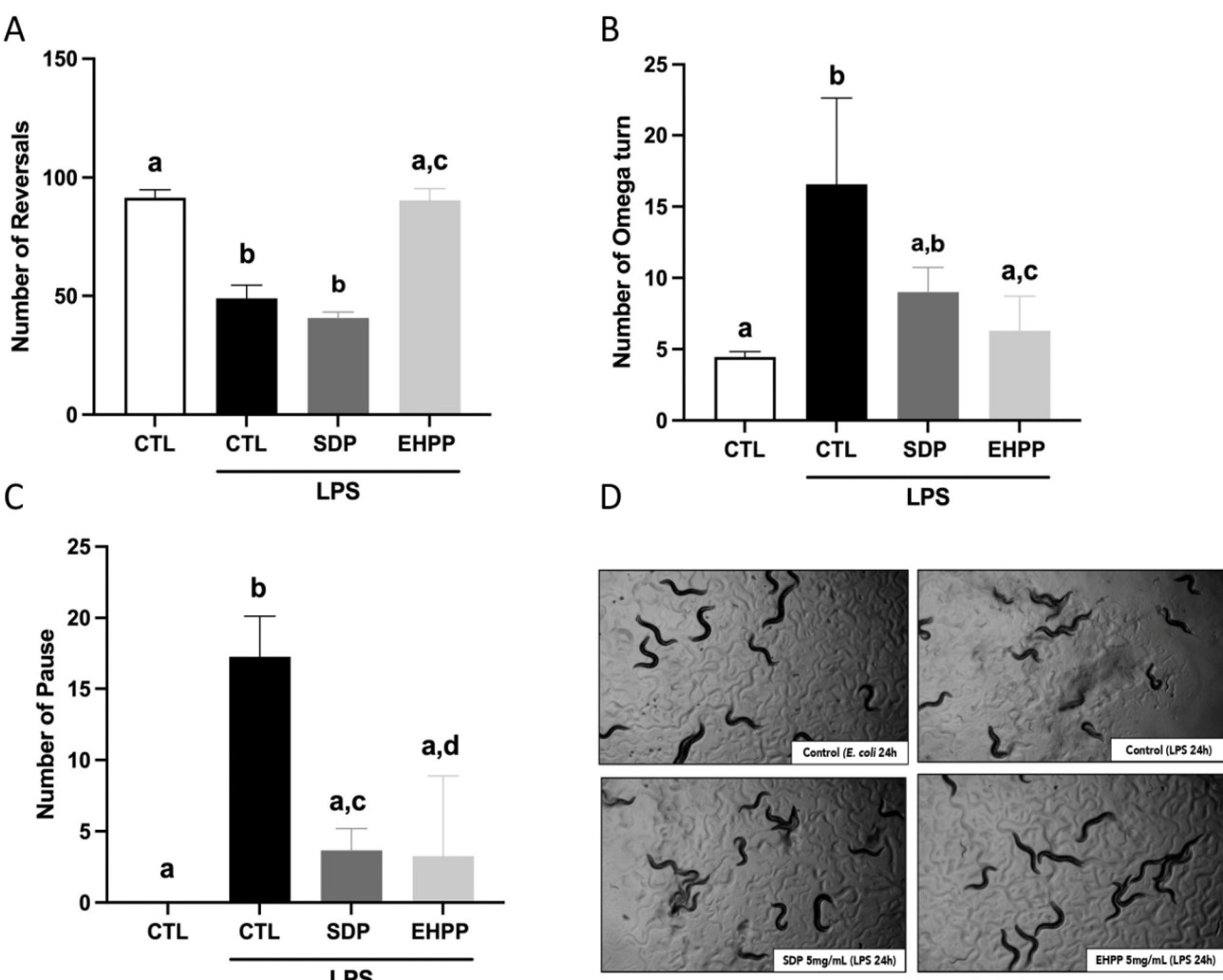

**Fig 6.** Summary of behavioral phenotypes after LPS injury in *C. elegans* Bristol N2 treated with EHPP or SDP at 5 mg/mL, number of Reversal (A), Omega turn (B), and Pause (C). Representative images from each group tested (D). Statistics: one-way ANOVA (Tukey test). Results are expressed as means ± SEM (n = 3, 60 worms in each group/replicate). Means with different letters are significantly different ($p < 0.05$). CTL: control; EHPP: Enzymatic hydrolyzed porcine plasma; SDP: Spray-dried plasma.

### Functional evaluation of EHPP in mice intestinal inflammation

**Body weight changes in SEB-induced intestinal inflammation.** The initial body weight of mice was similar between the different groups. Mice fed SDP or EHPP feeds showed a higher rate of body weight gain before SEB administration (both, $p < 0.05$, Fig 7A). The body weight gain before SEB injection was higher for mice fed the SDP or EHPP feeds compared with control mice ($p < 0.05$, Fig 7B). SEB administration reduced the body weight of control mice and those fed EHPP feed ($p < 0.05$, Fig 7C), while mice fed SDP feed presented intermediate values (S5 Table).

**Mesenteric lymph node lymphocytes in SEB-induced intestinal inflammation.** SEB administration did not modify the percentage of Th lymphocytes between the different groups (Fig 8A). SEB administration increased the percentage of activated Th lymphocytes ($p < 0.05$, Fig 8B). However, supplementation with SDP or EHPP reduced the percentage of activated Th lymphocytes in comparison with control mice ($p < 0.05$). SEB administration did not change

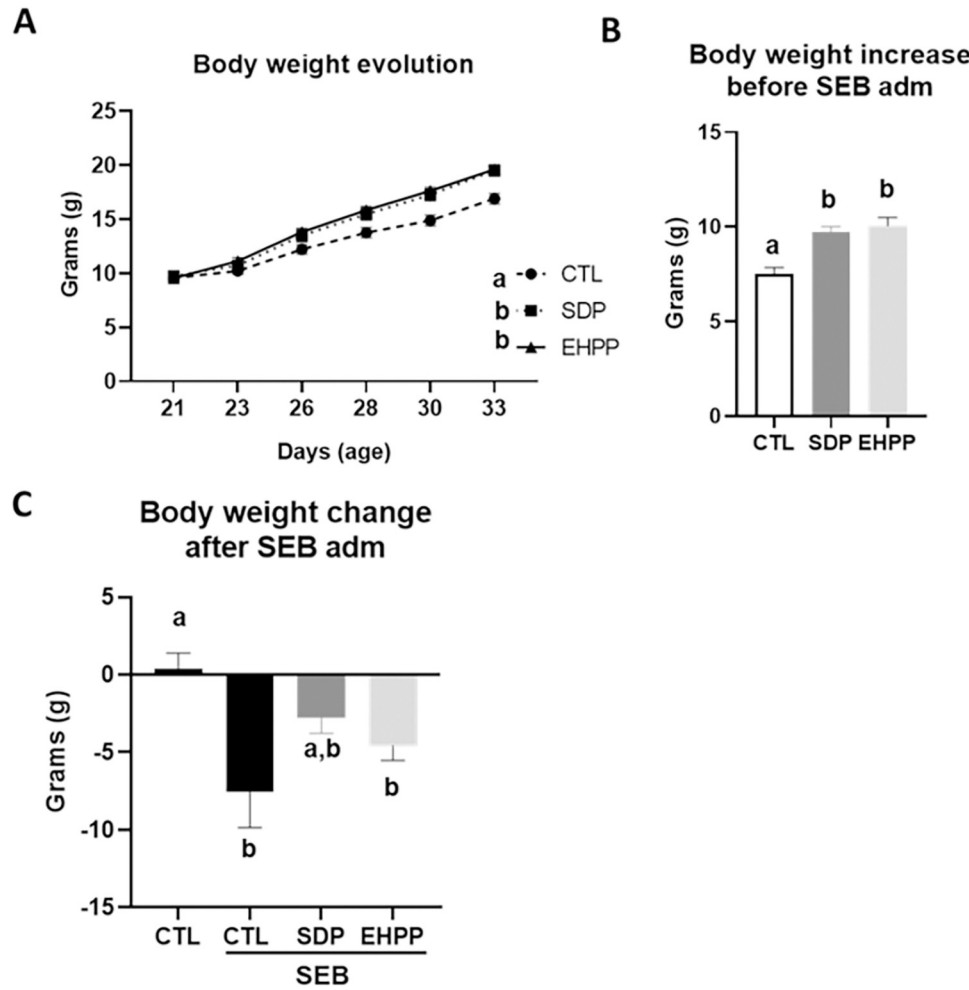

**Fig 7.** Body weight evolution before administration of SEB (A), body weight increase before administration of SEB (B) and body weight change after SEB administration (C). Results are expressed as means ± SEM (n = 9–13 mice). The results of body weight evolution were analyzed using area under the curve. Statistics: for body weight evolution, two-way ANOVA (feed and time; Fisher multiple comparison test); body weight change previous and after SEB administration, one-way ANOVA (Fisher multiple comparison test). Means with different letters are significantly different (p<0.05). CTL: control; EHPP: Enzymatic hydrolyzed porcine plasma; SEB: *S. aureus* enterotoxin B; SDP: Spray-dried plasma.

the percentage of regulatory Th lymphocytes (Fig 8C) in control mice, whereas mice fed SDP and EHPP feeds presented higher values of this lymphocyte population (p<0.05). Finally, the ratio between activated and regulatory Th lymphocytes increased in mice that received SEB, and both dietary supplements prevented this effect (p<0.05, Fig 8D and S6 Table).

**Cytokine expression in jejunum mucosa in SEB-induced intestinal inflammation.** SEB administration augmented the expression of *Tnf-α* in the jejunum mucosa (p<0.05, Fig 9A), which was completely prevented by both supplements (p<0.05). SEB administration stimulated the expression of *Ifn-γ* (p<0.05, Fig 9B). Supplementation with SDP did not significantly modify the effect of SEB, while supplementation with EHPP further increased the expression of this cytokine (p<0.05). The expression of anti-inflammatory cytokine *Il-10* did not change when mice received the SEB administration, but both supplements increased its expression (p<0.05, Fig 9C and S7 Table).

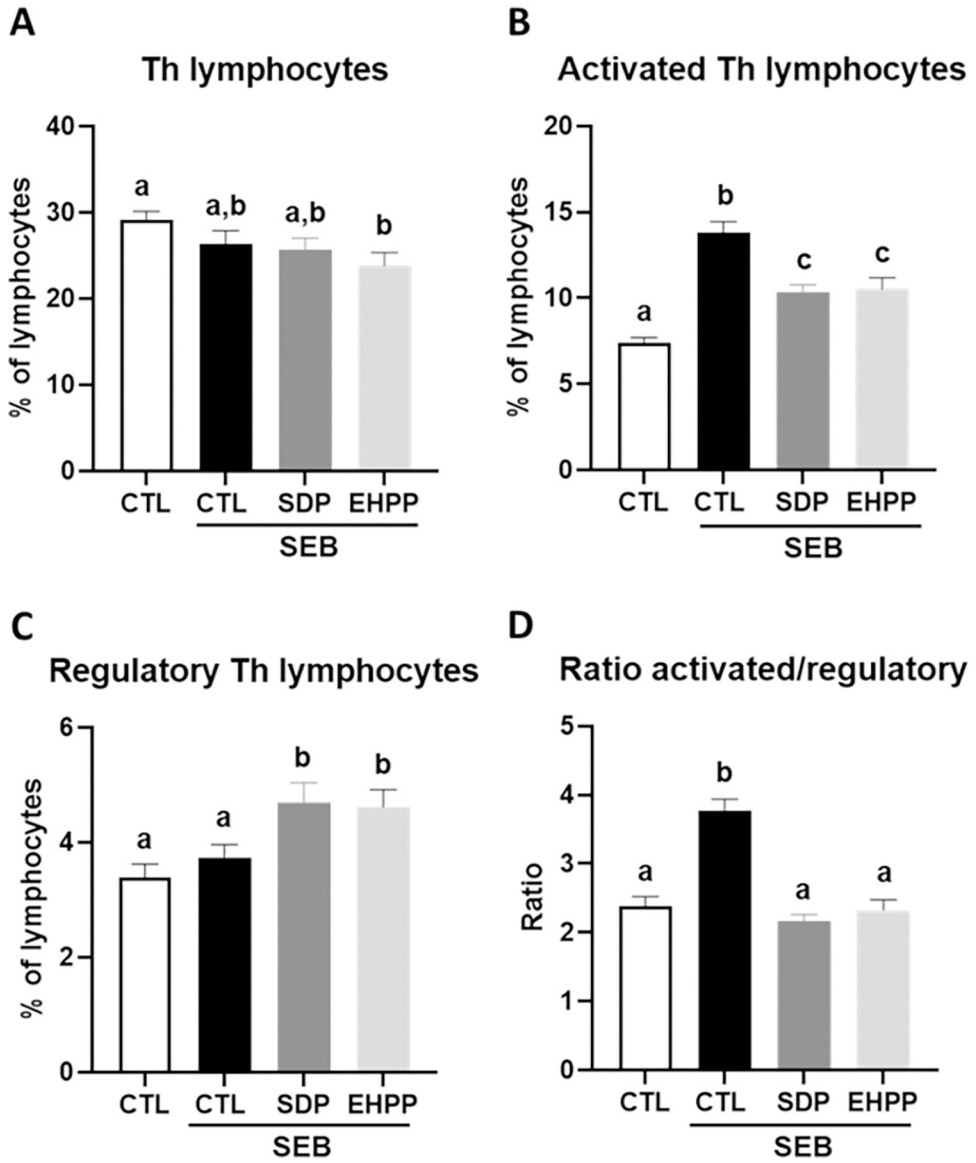

**Fig 8.** Percentages of Th lymphocytes (A), activated (act.) Th lymphocytes (B), regulatory (reg) Th lymphocytes (C), and the ratio between act and reg Th lymphocytes (D). Results are expressed as means ± SEM (n = 9–13 mice). Statistics: one-way ANOVA (Fisher multiple comparison test). Means without a common letter differ, $p < 0.05$. CTL: control; EHPP: Enzymatic hydrolyzed porcine plasma; SEB: *S. aureus* enterotoxin B; SDP: Spray-dried plasma.

**Body weight changes in *E. coli* infection.** Initial body weight was similar between the different experimental groups. Both SDP and EHPP supplemented feeds increased the rate of body weight gain before *E. coli* infection ($p < 0.05$, Fig 10A and 10B, S8 Table), although mice fed SDP feed showed higher values ($p < 0.05$). *E. coli* infection reduced the rate of body weight gain of control mice ($p < 0.05$, Fig 10C and S9 Table). EHPP feed, prevented this effect ($p < 0.05$).

**Cytokine expression in mesenteric lymph node leukocytes in *E. coli* infection.** *E. coli* infection increased the expression of pro-inflammatory cytokines (*Il-6* and *Il-1β*, all $p < 0.05$, Fig 11A and 11B) in leukocytes from MLN (control mice). The expression of both cytokines

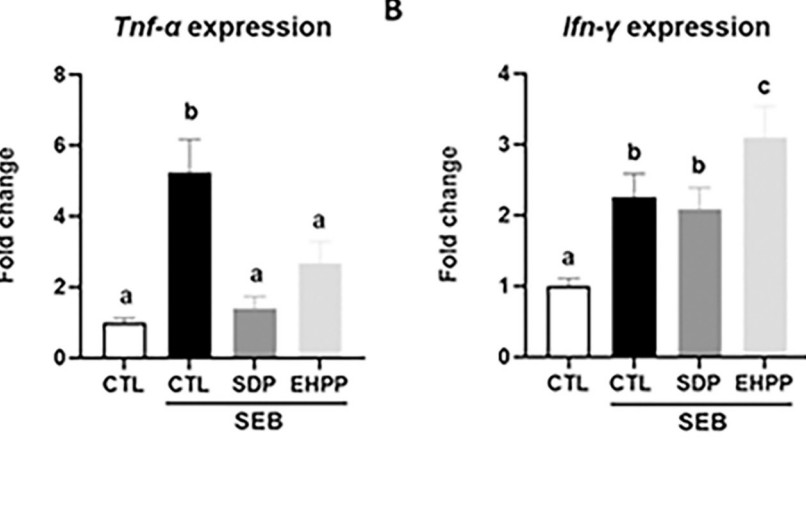

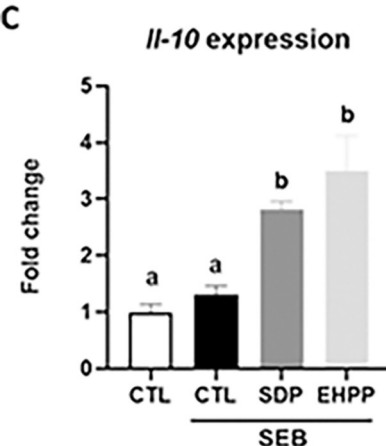

**Fig 9.** Expression of *Tnf-α* (A), *Ifn-γ* (B), and *Il-10* (C) in jejunum mucosa. Results are expressed as means ± SEM (n = 4–5 mice). Statistics: one-way ANOVA (Brown Forsythe and Welch test for *Tnf-α* expression; Fisher multiple comparison test for *Ifn-γ* and *Il-10* expression). Means without a common letter differ, $p < 0.05$. CTL: control; EHPP: Enzymatic hydrolyzed porcine plasma; SEB: *S. aureus* enterotoxin B; SDP: Spray-dried plasma.

was reduced in mice fed SDP feed ($p < 0.05$), and only a tendency of reduction in *Il-1β* was observed by EHPP supplementation (S10 Table).

## Discussion

In this study, the characteristics of an EHPP commercial product containing highly digestible peptides, free amino acids, and functional proteins, were determined. EHPP differs from SDP in ash, crude protein, solubility, and minerals. Although the level of free amino acids is higher in EHPP due to protein hydrolysis, the amino acid concentration profile is very similar to SDP.

In the European Union (EU) the use of animal protein is not permitted for ruminant feed. However, according to the EU regulation, the use of hydrolyzed protein in all farm animal feeds, including ruminant feed is allowed. Furthermore, in the case of non-ruminant hydrolyzed products, it is not even necessary to have all peptides below 10,000 $M_r$. So, the novel EHPP presented in this study meets all requirements for being used to supplement the feed of any farm animal species.

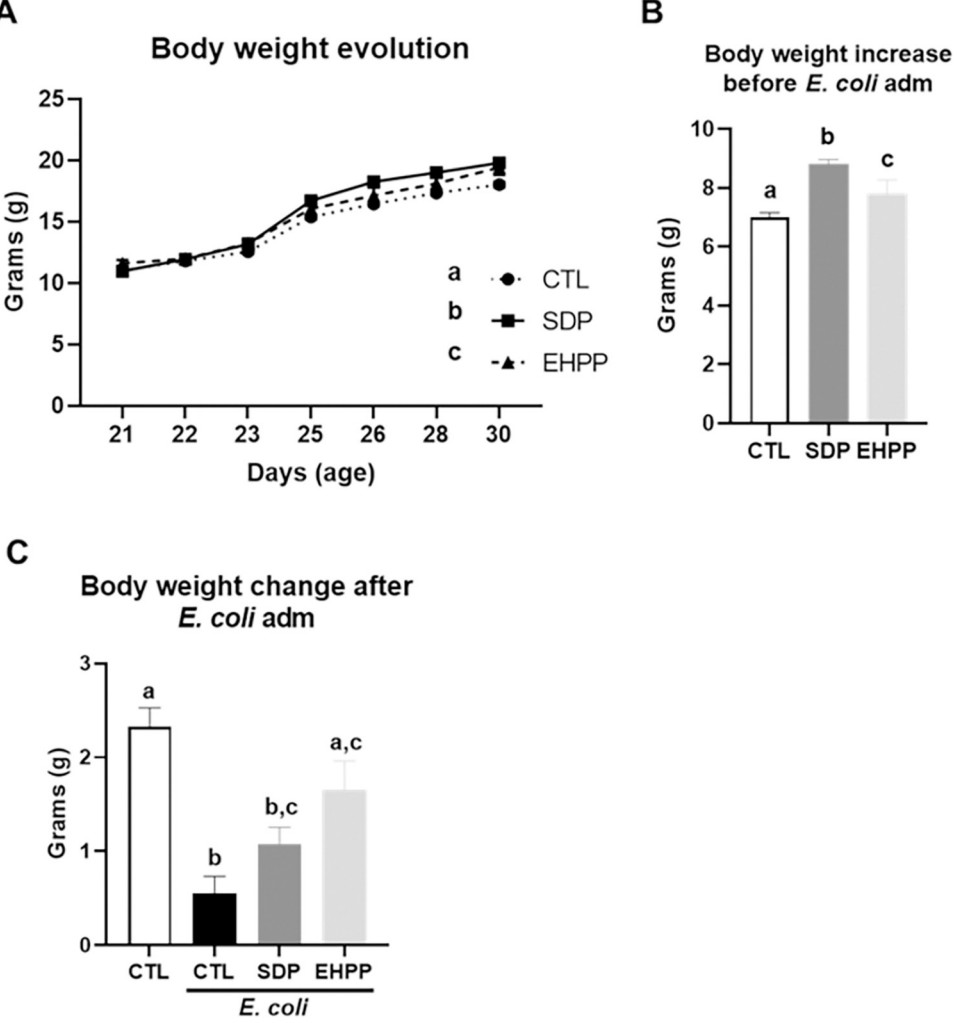

**Fig 10.** Body weight evolution previous administration of *E. coli* (A), body weight increase before administration of *E. coli* (B), and body weight change after *E. coli* administration (C). Results are expressed as means ± SEM (n = 9–24 mice). The results of body weight evolution were analyzed using the area under the curve. Statistics: for body weight evolution, two-way ANOVA (feed and time; Fisher multiple comparison test); body weight change previous and after SEB administration, one-way ANOVA (Fisher multiple comparison test). Means without a common letter differ, p<0.05. CTL: control; EHPP: Enzymatic hydrolyzed porcine plasma; SDP: Spray-dried plasma.

As expected, differences were found in the SEC chromatographic profiles between SDP and EHPP, suggesting changes in $M_r$ profile after the hydrolysis. EHPP did not contain proteins comprised between 100,000 and 1,000,000 $M_r$, whereas SDP contained proteins above 100,000 $M_r$ (57% of the protein relative abundance). A total of 73% of protein relative abundance of EHPP was below 8,000 $M_r$. MALDI-TOF-MS confirmed the effectiveness of hydrolysis with a simple and rapid characterization, but LC-MS/MS was necessary for a more detailed identification of peptides and their amino acid sequences. A total of 92 peptides in Fraction 2' of EHPP, which included peptides from 1,000 to 8,000 $M_r$, were identified after LC-MS/MS. According to the size of the sequences of the peptides found (around 7–47 amino acids with most of them below 20 residues), they may have a positive functionality, since it is known that most BAP are composed of 2–20 amino acid residues [20, 26, 27]. In addition, BAP with $M_r \leq$ 3,000 have been reported to present higher antioxidant properties and are easier to cross the

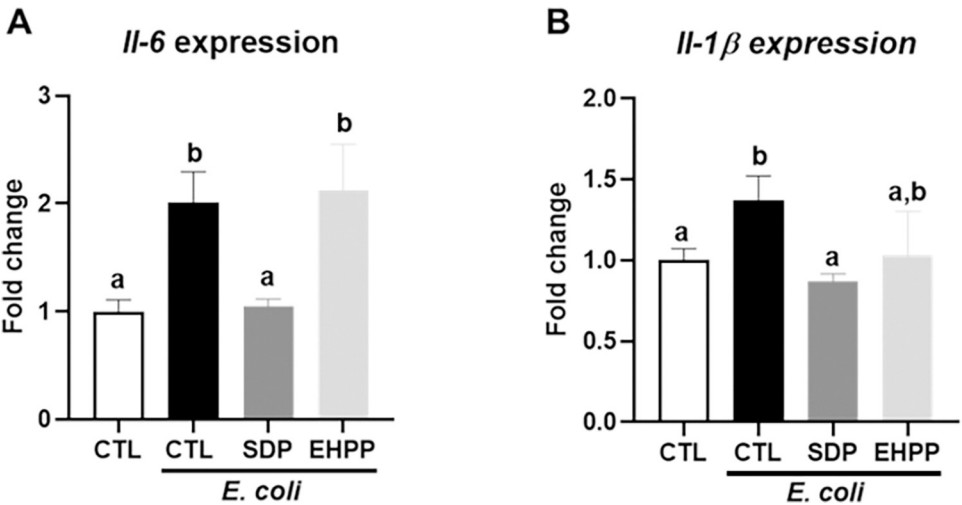

**Fig 11.** Expression of pro-inflammatory cytokines in mesenteric lymph nodes; (A) *Il-6* (A) expression and *Il-1β* (B) in mesenteric lymph nodes. Results are expressed as means ± SEM (n = 6–11 mice). Statistics: one-way ANOVA (Brown Forsythe and Welch test). Means without a common letter differ, p<0.05. CTL: control; EHPP: Enzymatic hydrolyzed porcine plasma; SDP: Spray-dried plasma.

intestinal barrier and exert the antioxidant effect, hence interacting more effectively with free radicals [40].

Past research studies described that antioxidative peptides usually contain hydrophobic amino acids, such as alanine (A), phenylalanine (F), glycine (G), leucine (L), isoleucine (I), methionine (M), proline (P), valine (V), and tryptophan (W) [39]. Those residues were found in all the 92 characterized peptide sequences, suggesting that they could enhance the scavenging of free radicals as has been previously reported. Among these hydrophobic amino acids, leucine (L) was found in 63 of these peptides. In addition, hydrophobic (Gly, Val, Leu, Pro, and Phe), negatively charged (Glu) and aromatic (Tyr) amino acids are the ones that mostly prevail in immunomodulatory peptides [41].

According to BIOPEP-UWM database of bioactive peptides (formerly BIOPEP, https://biochemia.uwm.edu.pl/en/biopep-uwm-2/), over 4,800 BAP have been reported to date, which are classified based on the bioactivity they exhibit, i.e., antimicrobial, antithrombotic, antihypertensive, immunomodulatory, opioid, mineral binding and antioxidants, among others. Such activities have been related to the prevention or treatment of different diseases, including cancer, immune disorders, and cardiovascular diseases [42]. Reviewing the sequences of the 92 peptides identified in Fraction 2', we found that only the sequence IIAPPER was previously described in the BIOPEP database. The peptide IIAPPER has been previously described from hydrolyzed insect (*Gryllodes sigillatus*), and its antioxidant activity has been evaluated by 2,2-diphenyl-1-picrylhydrazyl (DPPH) and [(2, 29-azinobis ((3-ethylene bezothiazoline) 6-sulphonic acid)] (ABTS) assays. In addition, this peptide has anti-inflammatory activity, which has been measured by lipoxygenase inhibitory activity (LOX) and cyclooxygenase 2 inhibitory activity (COX 2). Furthermore, this peptide sequence exhibits high ACE inhibitory capacity [43]. The other sequences found had not been previously described as BAP, suggesting that other non-yet determined BAP sequences may have relevant biological functions. These biological functions may be explained by considering the proteins that originated the BAP (Table 4).

Yang et al (2020) identified and synthesized seven novel peptides with antioxidant activity from duck plasma proteins. Among these peptides, the sequence EVGK exhibited the highest

$Fe^{2+}$ chelating ability [44]. In the peptide sequences identified in the present study, three peptides were found that included the sequence EVG, which may suggest that this amino acid sequence can be responsible of the antioxidant property of the peptide. Further gastrointestinal digestion can release this sequence from the three longer peptides found in Fraction 2'.

Kar et al. (2016) used an advanced proteomic and bioinformatic approach to characterize the protein component of 6 different protein sources, including SDP, and predicted the biofunctionalities of their *in vivo* digestion products according to the BIOPEP database. They found that SDP was the protein source with higher biofunctionality including: ACE inhibitor (67.37%), antiamnesic (0.35%), ion flow regulating peptides (2.48%), peptides with inhibition activity like dipeptidyl-aminopeptidase IV inhibitor, calmodulin-dependent cyclic nucleotide phosphodiesterase inhibitor, and renin inhibitor (0.71%), antioxidative peptides (18.79%), antithrombotic peptides (0.35%), hypotensive (1.77%), stimulating peptides like glucose uptake stimulating peptide and peptide stimulating vasoactive substance release (7.8%) and antibacterial peptides (0.34%) properties [12]. The fact that most peptides found in Fraction 2' of this study contain hydrophobic amino acids support the presence of antioxidant and ACE inhibitor properties of the BAP present in the EHPP product.

In all the *in vivo* models studied, EHPP partially retained the functionality observed for SDP. For instance, in a situation of oxidative stress in *C. elegans*, EHPP improved considerably the basal situation compared to the control, having a similar effect to SDP. Oxidative stress plays an important role in aging and several neurological disorders in animals and humans, such as Alzheimer's and Parkinson's diseases [45]. Therefore, antioxidant compounds play an important role in the prevention of free radical-induced tissue damage associated with age [20]. Reduced oxidative stress in nematodes supplemented with EHPP was probably related to the enhanced scavenging of free radicals associated to BAP with the presence of hydrophobic amino acids. In addition, EHPP considerably improved the functionality of SDP during an infection process. This suggests that EHPP, not only maintains the functions of SDP, but it could also be more active in certain stress or challenge situations. Furthermore, both EHPP and SDP significantly improved the survival of the nematodes. The nematode strain CL2006 has been considered as an animal model of aging-related deterioration, since it expresses human Aβ₁₋₄₂ under control of a muscle-specific promoter and exhibits age-dependent paralysis due to Aβ₁₋₄₂ aggregation [45]. In this model, supplementation with SDP or EHPP significantly improved locomotor capacity. In previous studies, using an accelerated age mice model, supplementation of SDP reduced brain oxidative stress, inflammation, and neuronal deterioration [37, 46, 47]. In the present study, both treatments ameliorated age-dependent paralysis, suggesting that the results obtained by SDP supplementation can be partially explained by the BAP obtained after the enzymatic hydrolysis.

Additionally, when *C. elegans* was exposed to LPS, EHPP and SDP showed a high anti-inflammatory effect. These results are relevant since, to the authors knowledge, this is the first time that the anti-inflammatory effect of plasma products is described in a nematode model, although SDP has been previously reported to reduce inflammation in different challenge studies conducted in animal models [9, 14, 15, 48–50]. However, it is of high relevance that BAP present in the EHPP were able to mitigate the negative effects of LPS exposure at a higher degree than SDP, suggesting that some of the BAP present in the EHPP had superior anti-inflammatory effects compared to SDP. As indicated previously, the immunomodulatory effects of BAP are related to the presence of hydrophobic amino acids such as Gly, Val, Leu, Pro and Phe that were present in 42, 47, 63, 54 and 25 of the bioactive peptides identified in EHPP, respectively. In addition, other immunomodulatory peptides contain negatively charged amino acids like Glu and aromatic amino acids like Tyr, which were present in 58 and 27 of the peptide sequences in EHPP in Fraction 2' of EHPP. This may explain the higher anti-

inflammatory effect of EHPP. The use of the *C. elegans* model provides a rapid, simple, and efficient *in vivo* anti-inflammatory assay of novel functional ingredients based on plasma products, which until now were performed using higher order animal models, such as mice.

Both models of mild intestinal inflammation in mice were induced by intraperitoneal administration of SEB or by an oral challenge with *E. coli*. SEB belongs to the family of superantigens, which are potent polyclonal activators of the immune system that can activate a high percentage of T cells [51]. Dietary supplementation with SDP attenuates gut-associated lymphoid tissue (GALT) activation, reduces mucosal inflammation, and thereby preserves the intestinal barrier integrity in weanling rodents challenged with SEB [2, 36, 48, 52]. Finally, *E. coli* is an important cause of intestinal inflammation and diarrhea in humans and farm animals [53]. Enteric pathogens and their enterotoxins induce an inflammatory response and alter intestinal functions, such as epithelial permeability and transport, contributing to diarrhea [54]. Moreover, it can promote asymptomatic colonization, which causes chronic intestinal inflammation [55]. As commented early, the combination of these two mice models allows obtaining an accurate and broad insight into EHPP functionality, disclosing the advantages of EHPP and SDP.

In mice, both supplements were found to promote similar body weight gain in non-challenged animals, higher than those fed the control feed. In fact, similar results have been observed for both supplements in farm animals [56, 57]. This effect is attributable to the fact that these supplements can modulate the immune response and improve the integrity of the intestinal mucosa, allowing greater availability of nutrients for growth [58, 59]. Furthermore, during intestinal inflammation, animals fed the supplemented feed showed less weight loss than those fed the control feed, indicating a lower magnitude of the inflammatory process, as intestinal inflammation results in lower feed intake, weight loss, nutritional depletion, and gastrointestinal dysfunction [60].

In terms of modulation of the intestinal immune response, both supplements showed similar effects on MLN lymphocyte populations following SEB challenge, confirming previous results with this model [2] and in line with previous observations in a colitis mice model [9] and in mice with acute lung inflammation [14]. EHPP and SDP supplementation reduced the percentage of activated Th lymphocytes and increased the regulatory ones. This effect on the ratio between activated and regulatory T cells paralleled changes in the expression of pro- and anti-inflammatory cytokines. Indeed, both EHPP and SDP reduced the expression of the pro-inflammatory cytokine *Tnf-α* and promoted the expression of the anti-inflammatory *Il-10* in the mucosa of jejunum, reducing the intensity of the intestinal inflammatory response. Indeed, the main role of regulatory T lymphocytes is maintaining immune homeostasis and suppressing intestinal inflammation that results from aberrant immune responses to self-antigens and commensal bacteria [61]. These highly similar responses of both supplements in some variables would indicate that they share some BAP that are eventually involved in the regulation of the mucosal immune system.

However, for some immune variables analyzed, the effects of EHPP and SDP differed significantly. For example, SDP did not modify the effects of SEB on *Ifn-γ* expression, consistent with previous studies with this model [2], while supplementation with EHPP increased its expression. This indicates that EHPP drives the mucosal immune response towards a Th1-type response (cell-mediated immunity), as *Ifn-γ* enhances the activation and proliferation of Th1 cells [62, 63]. Moreover, both supplements also differed on the immune response to *E. coli* challenge, as SDP prevented its effects on the expression of *Il-1β* and *Il-6* in MLN leukocytes, while EHPP did not change the *Il-6* expression, suggesting that the products may be acting through different mechanisms. In this sense, the mechanism of action of SDP could involve distinct elements as it contains growth factors, cytokines, immunoglobulins, and other

biologically active compounds [2, 64, 65]. In contrast, in EHPP these proteins are hydrolyzed and, although the peptides generated may have biological activity, the mechanism of action is likely to be different from that of the protein from which they originate. Specifically, SDP and EHPP differ in mineral profiles, including differences in molybdenum and zinc concentrations, critical elements for maintaining various animal biochemical and physiological functions [66]. In addition, SDP has also shown to have prebiotic effects on gut bacteria in mice [19] and pigs [18], which may also contribute to its mechanism of action. In contrast, it is unknown if EHPP affects gut microbiota. Further studies are needed to better understand and distinguish if EHPP feed has direct effects on immune modulation or if it has indirect effects through modulation of the microbiota.

Only a few studies showing the benefits of EHPP supplementation in farm animals have been described to date. In a study conducted with weaned pigs [57], the supplementation with 5.6% of EHPP in a feed fed during the first 14 days post-weaning increased body weight, average daily gain, daily feed intake, gain to feed ratio, and decreased the incidence of diarrhea compared with a control feed. A recent study using gilthead sea bream (*Sparus aurata*) showed that EHPP included in low fishmeal feeds promoted growth and feed efficiency, as well as enhanced the immune response, which indicates that this is a safe and functional ingredient [67]. However, further studies are needed in different species, such as calves and poultry, to confirm its potential beneficial effects of EHPP.

In conclusion, the present study fully characterized the nutritional and molecular properties of EHPP and demonstrated, using three different *in vivo* models subjected to different challenge conditions, that it provides health-promoting benefits that may be due to the antioxidant and immunomodulatory effects of its containing BAP.

## Supporting information

**S1 Table. Food clearance assay results in *C. elegans* to evaluate the toxicity of SDP and EHPP at different concentrations for 6 days.**
(XLSX)

**S2 Table. Oxidative stress response of different *C. elegans* after treatment with EHPP and SDP supplementation at 5 mg/mL.**
(XLSX)

**S3 Table. Age-dependent paralysis in *C. elegans* CL2006 after treatment with EHPP and SDP supplementation at 5 mg/mL.**
(XLSX)

**S4 Table. Behavioral phenotypes after LPS injury in *C. elegans* after treatment with EHPP and SDP supplementation at 5 mg/mL.**
(XLSX)

**S5 Table. Body weight results at different days during the SEB challenge study in mice.**
(XLSX)

**S6 Table. Lymphocytes results of different groups during the SEB challenge study in mice.**
(XLSX)

**S7 Table. Cytokines expression (Tnf-α, Ifn-γ and Il-10) results in jejunum mucosa of different groups during the SEB challenge study in mice.**
(XLSX)

**S8 Table. Body weight results at different days previous to the *E. coli* challenge in mice.**
(XLSX)

**S9 Table. Body weight results at different days after the *E. coli* challenge in mice.**
(XLSX)

**S10 Table. Cytokines expression (Il-6 and Il-1β) results in jejunum mucosa of different groups during the *E. coli* challenge study in mice.**
(XLSX)

## Acknowledgments

The authors would like to acknowledge all the staff that was involved in the animal housing facilities of Facultat de Farmàcia i Ciències de l'Alimentació and Institut de Nutrició i Seguretat Alimentària of Universitat de Barcelona.

## Author Contributions

**Conceptualization:** Marc Solà-Ginés, Lluïsa Miró, Christian Griñán-Ferré, Mercè Pallàs, Anna Pérez-Bosque, Miquel Moretó, Laura Pont, Fernando Benavente, José Barbosa, Carmen Rodríguez, Javier Polo.

**Formal analysis:** Marc Solà-Ginés, Lluïsa Miró, Aina Bellver-Sanchis, Christian Griñán-Ferré, Anna Pérez-Bosque, Laura Pont, Carmen Rodríguez, Javier Polo.

**Investigation:** Marc Solà-Ginés, Lluïsa Miró, Aina Bellver-Sanchis, Christian Griñán-Ferré, Anna Pérez-Bosque, Laura Pont, Carmen Rodríguez, Javier Polo.

**Methodology:** Marc Solà-Ginés, Lluïsa Miró, Aina Bellver-Sanchis, Christian Griñán-Ferré, Anna Pérez-Bosque, Laura Pont, Carmen Rodríguez, Javier Polo.

**Writing – original draft:** Marc Solà-Ginés, Lluïsa Miró, Christian Griñán-Ferré, Anna Pérez-Bosque, Laura Pont, Fernando Benavente, Carmen Rodríguez, Javier Polo.

**Writing – review & editing:** Marc Solà-Ginés, Lluïsa Miró, Christian Griñán-Ferré, Anna Pérez-Bosque, Miquel Moretó, Laura Pont, Fernando Benavente, José Barbosa, Carmen Rodríguez, Javier Polo.

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
