## [Decision Letter · Decision Letter 0]

16 Jan 2024

PONE-D-23-36193Nutritional, molecular and functional properties of a novel enzymatically hydrolyzed porcine plasma productPLOS ONE

Dear Dr. Solà Ginés,

Thank you for submitting your manuscript to PLOS ONE. After careful consideration, we feel that it has merit but does not fully meet PLOS ONE’s publication criteria as it currently stands. Therefore, we invite you to submit a revised version of the manuscript that addresses the points raised during the review process.

We look forward to receiving your revised manuscript.

Kind regards,

Mahmoud A.O. Dawood, PhD

Academic Editor

PLOS ONE

“Funding for this study was provided by APC Europe, S.L.U., Granollers, Spain that is a company that manufacture animal blood products for animal consumption. The company provided support in the form of salaries for authors M.S.-G., C.R., and J.P. retrospectively, but the company did not have any additional role in the study design, data collection and analysis, decision to publish, or preparation of the manuscript. The specific roles of these authors are articulated in the ‘author contributions’ section. In addition, the authors would like to acknowledge the Centro para el Desarrollo Tecnológico Industrial (CDTI project IDI-20180886) for co-funding this work.”

“The authors would like to acknowledge the Centro para el Desarrollo Tecnológico Industrial (CDTI project IDI-20180886) for co-funding this work.”

“Funding for this study was provided by APC Europe, S.L.U., Granollers, Spain that is a company that manufacture animal blood products for animal consumption. The company provided support in the form of salaries for authors M.S.-G., C.R., and J.P. retrospectively, but the company did not have any additional role in the study design, data collection and analysis, decision to publish, or preparation of the manuscript. The specific roles of these authors are articulated in the ‘author contributions’ section. In addition, the authors would like to acknowledge the Centro para el Desarrollo Tecnológico Industrial (CDTI project IDI-20180886) for co-funding this work.”

5. We note that your Data Availability Statement is currently as follows: [All relevant data are within the manuscript and its Supporting Information files.]

6. PLOS requires an ORCID iD for the corresponding author in Editorial Manager on papers submitted after December 6th, 2016. Please ensure that you have an ORCID iD and that it is validated in Editorial Manager. To do this, go to ‘Update my Information’ (in the upper left-hand corner of the main menu), and click on the Fetch/Validate link next to the ORCID field. This will take you to the ORCID site and allow you to create a new iD or authenticate a pre-existing iD in Editorial Manager. Please see the following video for instructions on linking an ORCID iD to your Editorial Manager account: https://www.youtube.com/watch?v=_xcclfuvtxQ.

7. We note that you have included the phrase “data not shown” in your manuscript. Unfortunately, this does not meet our data sharing requirements. PLOS does not permit references to inaccessible data. We require that authors provide all relevant data within the paper, Supporting Information files, or in an acceptable, public repository. Please add a citation to support this phrase or upload the data that corresponds with these findings to a stable repository (such as Figshare or Dryad) and provide and URLs, DOIs, or accession numbers that may be used to access these data. Or, if the data are not a core part of the research being presented in your study, we ask that you remove the phrase that refers to these data.

Reviewers' comments:

Reviewer's Responses to Questions

**Comments to the Author**

1. Is the manuscript technically sound, and do the data support the conclusions?

Reviewer #1: Yes

Reviewer #2: Yes

2. Has the statistical analysis been performed appropriately and rigorously? 

Reviewer #1: Yes

Reviewer #2: Yes

3. Have the authors made all data underlying the findings in their manuscript fully available?

Reviewer #1: Yes

Reviewer #2: Yes

4. Is the manuscript presented in an intelligible fashion and written in standard English?

Reviewer #1: Yes

Reviewer #2: Yes

5. Review Comments to the Author

Reviewer #1: The authors use a sound methodology and carry out the experiments with precision. The manuscript is well written and easy to follow. The results are scientifically sound and of high quality, the methods are appropriate, and the assays were well performed.

The in vivo experiments in C. elegans and mouse models of intestinal inflammation provide compelling evidence for the anti-inflammatory effects of EHPP, showing improvements in survival, motility, and attenuation of pro-inflammatory responses.

Overall, this study shows that EHPP is a promising candidate for health benefits, possibly due to its characteristic low molecular weight bioactive peptides, that show potential as immunomodulatory and antioxidant agents.

There's a particular point to highlight about bioactive peptides (BAPs) in lines 82 to 89. These peptides aren't just components of food or inactive parts of proteins in their original molecules. They can also be found on or produced by microorganisms, plants, or animals. Therefore, I recommend that the authors expand on BAP with a more comprehensive explanation in the Introduction. Therefore, I suggest that the authors include a more detailed description of BAP in the Introduction.

The comparison between EHPP and SDP shows differences in their mechanisms of action, especially in the markers IL-6 and TNF-alpha, suggesting different pathways for their beneficial effects. Do the authors have an explanation for this difference?

A paper comparing the nutritional composition of SDP and EHPP was published in December 2023, https://doi.org/10.3390/molecules28237917. I suggest that the authors compare their results with those presented in this paper.

Reviewer #2: The manuscript provides valuable insights into the nutritional and molecular properties of EHPP. However, I have some specific comments on certain sections of the manuscript. I trust that the authors will carefully consider my suggestions and make appropriate improvements.

Major Problem:

One significant concern that requires attention is the lack of clarity in the figures. While I trust the authors' interpretation of the results, the figures need improvement as they are currently unreadable.

Minor Comments and Questions:

Lines 109-114: The detailed information provided in this section appears excessive and may be more appropriately placed elsewhere in the text, perhaps in the methodology or discussion. Consider relocating this content to maintain a smoother flow in the presentation.

Lines 123-124: The statement "Crude fiber (CF) was used as an indicator of fiber digestion" requires clarification. Please explain the intended meaning.

Lines 124-125: The phrase implies that tryptophan is not an amino acid. Revise the sentence for clarity.

Methodology section: It is recommended to avoid the use of the first person in describing methods. Convert sentences to the impersonal form for a more formal tone and structure.

Table 1: Why was less methionine used in the control feed? Additionally, why was less lysine used in the EHPP diet when Table 2 indicates it has a lower tyrosine content than SDP?

Line 357: Is it appropriate to categorize the fat content as "very low"?

Discussion section: At times, navigating through the discussion becomes challenging in distinguishing between studies by other authors and the current results. Emphasizing when discussing the present findings versus referencing other studies would enhance clarity for readers.

Addressing the major concern regarding figure clarity and refining minor details will contribute to the overall improvement of the manuscript.

6. PLOS authors have the option to publish the peer review history of their article (what does this mean?). If published, this will include your full peer review and any attached files.

Reviewer #1: No

Reviewer #2: No

---

## [Author Response · Author response to Decision Letter 0]

11 Mar 2024

The entire Manuscript has been revised and if there are no mistakes, all changes were performed following the PLOS ONE requirements.

We appreciate the suggestion, but we already provided all data in the manuscript.

“Funding for this study was provided by APC Europe, S.L.U., Granollers, Spain that is a company that manufacture animal blood products for animal consumption. The company provided support in the form of salaries for authors M.S.-G., C.R., and J.P. retrospectively, but the company did not have any additional role in the study design, data collection and analysis, decision to publish, or preparation of the manuscript. The specific roles of these authors are articulated in the ‘author contributions’ section. In addition, the authors would like to acknowledge the Centro para el Desarrollo Tecnológico Industrial (CDTI project IDI-20180886) for co-funding this work.”

Funding is updated in the reviewed Manuscript. In addition, a statement about the role of the funders was added.

“The authors would like to acknowledge the Centro para el Desarrollo Tecnológico Industrial (CDTI project IDI-20180886) for co-funding this work.”

“Funding for this study was provided by APC Europe, S.L.U., Granollers, Spain that is a company that manufacture animal blood products for animal consumption. The company provided support in the form of salaries for authors M.S.-G., C.R., and J.P. retrospectively, but the company did not have any additional role in the study design, data collection and analysis, decision to publish, or preparation of the manuscript. The specific roles of these authors are articulated in the ‘author contributions’ section. In addition, the authors would like to acknowledge the Centro para el Desarrollo Tecnológico Industrial (CDTI project IDI-20180886) for co-funding this work.”

This Funding Statement is correct, since it already has the addition of “In addition, the authors would like to acknowledge the Centro para el Desarrollo Tecnológico Industrial (CDTI project IDI-20180886) for co-funding this work”. Moreover, we added a new sentence in the acknowledgements section.

5. We note that your Data Availability Statement is currently as follows: [All relevant data are within the manuscript and its Supporting Information files.]

We added ten supplementary tables containing the raw data requested regarding to figures 3 to 11. 

6. PLOS requires an ORCID iD for the corresponding author in Editorial Manager on papers submitted after December 6th, 2016. Please ensure that you have an ORCID iD and that it is validated in Editorial Manager. To do this, go to ‘Update my Information’ (in the upper left-hand corner of the main menu), and click on the Fetch/Validate link next to the ORCID field. This will take you to the ORCID site and allow you to create a new iD or authenticate a pre-existing iD in Editorial Manager. Please see the following video for instructions on linking an ORCID iD to your Editorial Manager account: https://www.youtube.com/watch?v=_xcclfuvtxQ. 

MS-G just signed into ORCID and information was validated in Editorial Manager as required.

7. We note that you have included the phrase “data not shown” in your manuscript. Unfortunately, this does not meet our data sharing requirements. PLOS does not permit references to inaccessible data. We require that authors provide all relevant data within the paper, Supporting Information files, or in an acceptable, public repository. Please add a citation to support this phrase or upload the data that corresponds with these findings to a stable repository (such as Figshare or Dryad) and provide and URLs, DOIs, or accession numbers that may be used to access these data. Or, if the data are not a core part of the research being presented in your study, we ask that you remove the phrase that refers to these data. 

The sentence “data not shown” was remove as suggested. 

Reviewers' comments:

Reviewer's Responses to Questions

Comments to the Author

1. Is the manuscript technically sound, and do the data support the conclusions?

Reviewer #1: Yes

Reviewer #2: Yes

2. Has the statistical analysis been performed appropriately and rigorously? 

Reviewer #1: Yes

Reviewer #2: Yes

3. Have the authors made all data underlying the findings in their manuscript fully available?

Reviewer #1: Yes

Reviewer #2: Yes

4. Is the manuscript presented in an intelligible fashion and written in standard English?

Reviewer #1: Yes

Reviewer #2: Yes

5. Review Comments to the Author

Reviewer #1: The authors use a sound methodology and carry out the experiments with precision. The manuscript is well written and easy to follow. The results are scientifically sound and of high quality, the methods are appropriate, and the assays were well performed.

The in vivo experiments in C. elegans and mouse models of intestinal inflammation provide compelling evidence for the anti-inflammatory effects of EHPP, showing improvements in survival, motility, and attenuation of pro-inflammatory responses.

Overall, this study shows that EHPP is a promising candidate for health benefits, possibly due to its characteristic low molecular weight bioactive peptides, that show potential as immunomodulatory and antioxidant agents.

There's a particular point to highlight about bioactive peptides (BAPs) in lines 82 to 89. These peptides aren't just components of food or inactive parts of proteins in their original molecules. They can also be found on or produced by microorganisms, plants, or animals. Therefore, I recommend that the authors expand on BAP with a more comprehensive explanation in the Introduction. Therefore, I suggest that the authors include a more detailed description of BAP in the Introduction. As suggested by the reviewer, a couple of sentences and an extra paragraph detailing more the description of BAP were added in the Introduction.

The comparison between EHPP and SDP shows differences in their mechanisms of action, especially in the markers IL-6 and TNF-alpha, suggesting different pathways for their beneficial effects. Do the authors have an explanation for this difference?

A paper comparing the nutritional composition of SDP and EHPP was published in December 2023, https://doi.org/10.3390/molecules28237917. I suggest that the authors compare their results with those presented in this paper. A sentence was added almost at the end of the Discussion when comparing SDP and EHPP, citing and discussing some of the results from the article published in December 2023.

Reviewer #2: The manuscript provides valuable insights into the nutritional and molecular properties of EHPP. However, I have some specific comments on certain sections of the manuscript. I trust that the authors will carefully consider my suggestions and make appropriate improvements.

Major Problem:

One significant concern that requires attention is the lack of clarity in the figures. While I trust the authors' interpretation of the results, the figures need improvement as they are currently unreadable. From Figure 7 to 11 the resolution was improved, whereas from Figure 4 to 6 everything was changed to unify the figure style and to make them more understandable. We believe that now is easier to follow the Results.

Minor Comments and Questions:

Lines 109-114: The detailed information provided in this section appears excessive and may be more appropriately placed elsewhere in the text, perhaps in the methodology or discussion. Consider relocating this content to maintain a smoother flow in the presentation. Most part of the paragraph was modified, but the sentence of C. elegans did not fit anywhere else. The detailed information about the models in mice was eliminated, due to it is already mentioned in Materials and Methods.

Lines 123-124: The statement "Crude fiber (CF) was used as an indicator of fiber digestion" requires clarification. Please explain the intended meaning. We appreciated that the reviewer catches up this wrong statement. This was a mistake, and the reviewer is completely right that this sentence does not make sense. We changed this sentence in the revised version of the manuscript.

Lines 124-125: The phrase implies that tryptophan is not an amino acid. Revise the sentence for clarity. The reviewer is right, the sentence was a bit confusing. We think that now is much clearer.

Methodology section: It is recommended to avoid the use of the first person in describing methods. Convert sentences to the impersonal form for a more formal tone and structure. The entire Manuscript was reviewed, and all the sentences found using the first person were changed.

Table 1: Why was less methionine used in the control feed? Additionally, why was less lysine used in the EHPP diet when Table 2 indicates it has a lower tyrosine content than SDP? We really appreciated that the reviewer points out the incongruencies of the original Table 1. In fact, we found a major mistake in Table 1 for ingredients in diets with SDP and EHPP. We added the right values in the revised Table 1. The amount of methionine is higher in diets with SDP and EHPP compared to the control diet due to the low levels of methionine in SDP and EHPP compared to whey protein concentrate (see Table 3 for methionine values in SDP and EHPP). The indication of lysine was a mistake in the original table because the diets were not supplemented with lysine and the values that originally appeared in Table 1 were the lysine values of the diets (14.9 g/kg for control and SDP diet and 14.5 g/kg for the EHPP diets, indicating a similar nutritional composition). We modified Table 1 accordingly, adding the nutritional values of the diets, as we consider that this can be a useful information for the readers.

Line 357: Is it appropriate to categorize the fat content as "very low"? The reviewer is right and the value of fat at 2.6 and 2.4% cannot be considered very low. Following reviewer suggestion, we modify this sentence as follows “Crude fat was similar for SDP and EHPP (2.6% and 2.4% respectively) and crude fiber was very low for both products” in the revised version of the Manuscript.

Discussion section: At times, navigating through the discussion becomes challenging in distinguishing between studies by other authors and the current res

---

## [Decision Letter · Decision Letter 1]

18 Mar 2024

Nutritional, molecular and functional properties of a novel enzymatically hydrolyzed porcine plasma product

PONE-D-23-36193R1

Dear Dr. Solà Ginés,

We’re pleased to inform you that your manuscript has been judged scientifically suitable for publication and will be formally accepted for publication once it meets all outstanding technical requirements.

Kind regards,

Mahmoud A.O. Dawood, PhD

Academic Editor

PLOS ONE

Additional Editor Comments (optional):

Reviewers' comments:

Reviewer's Responses to Questions

**Comments to the Author**

1. If the authors have adequately addressed your comments raised in a previous round of review and you feel that this manuscript is now acceptable for publication, you may indicate that here to bypass the “Comments to the Author” section, enter your conflict of interest statement in the “Confidential to Editor” section, and submit your "Accept" recommendation.

Reviewer #1: All comments have been addressed

Reviewer #2: All comments have been addressed

2. Is the manuscript technically sound, and do the data support the conclusions?

Reviewer #1: Yes

Reviewer #2: Yes

3. Has the statistical analysis been performed appropriately and rigorously? 

Reviewer #1: Yes

Reviewer #2: Yes

4. Have the authors made all data underlying the findings in their manuscript fully available?

Reviewer #1: Yes

Reviewer #2: Yes

5. Is the manuscript presented in an intelligible fashion and written in standard English?

Reviewer #1: Yes

Reviewer #2: Yes

6. Review Comments to the Author

Reviewer #1: The authors have adequately addressed my comments. Thus, I have come to the conclusion that this manuscript is now considered suitable for publication.

Reviewer #2: (No Response)

7. PLOS authors have the option to publish the peer review history of their article (what does this mean?). If published, this will include your full peer review and any attached files.

Reviewer #1: No

Reviewer #2: **Yes: **Katarzyna Kazimierska

---

## [Editor Report · Acceptance letter]

29 Apr 2024

PONE-D-23-36193R1 

PLOS ONE

Dear Dr. Solà Ginés, 

I'm pleased to inform you that your manuscript has been deemed suitable for publication in PLOS ONE. Congratulations! Your manuscript is now being handed over to our production team.

Kind regards, 

on behalf of

Dr. Mahmoud A.O. Dawood 

Academic Editor

PLOS ONE